# Variant mutation G215C in SARS-CoV-2 nucleocapsid enhances viral infection via altered genomic encapsidation

Hannah C. Kubinski[1,☯,¤a], Hannah W. Despres[1,☯,¤b], Bryan A. Johnson[2,3,4],
Madaline M. Schmidt[1,¤c], Sara A. Jaffrani[1], Allyson H. Turner[1], Conor D. Fanuele[1],
Margaret G. Mills[5], Kumari G. Lokugamage[2], Caroline M. Dumas[6], David J. Shirley[7],
Leah K. Estes[2], Andrew Pekosz[8], Jessica W. Crothers[9], Pavitra Roychoudhury[5],
Alexander L. Greninger[5,10], Keith R. Jerome[5,10], Bruno Martorelli Di Genova[1],
David H. Walker[2,3,4,11], Bryan A. Ballif[6], Mark S. Ladinsky[12], Pamela J. Bjorkman[12],
Vineet D. Menachery[2,13,14,15], Emily A. Bruce[ID][1]*

1 Department of Microbiology and Molecular Genetics, Robert Larner, M.D. College of Medicine, University of Vermont, Burlington, Vermont, United States of America, 2 Department of Microbiology and Immunology, University of Texas Medical Branch, Galveston, Texas, United States of America, 3 Institute for Human Infection and Immunity, University of Texas Medical Branch, Galveston, Texas, United States of America, 4 Center for Tropical Diseases, University of Texas Medical Branch, Galveston, Texas, United States of America, 5 Virology Division, Department of Laboratory Medicine and Pathology, University of Washington, Seattle, Washington, United States of America, 6 Department of Biology, University of Vermont, Burlington, Vermont, United States of America, 7 Faraday, Inc. Data Science Department, Burlington, Vermont, United States of America, 8 W. Harry Feinstone Department of Molecular Microbiology and Immunology, The Johns Hopkins Bloomberg School of Public Health, Baltimore, Maryland, United States of America, 9 Department of Pathology and Laboratory Medicine, Robert Larner, MD College of Medicine, University of Vermont, Burlington, Vermont, United States of America, 10 Vaccine and Infectious Disease Division, Fred Hutchinson Cancer Research Center, Seattle Washington, United States of America, 11 Department of Pathology, University of Texas Medical Branch, Galveston, Texas, United States of America, 12 Division of Biology and Biological Engineering, California Institute of Technology, Pasadena, California, United States of America, 13 World Reference Center of Emerging Viruses and Arboviruses, University of Texas Medical Branch, Galveston, Texas, United States of America, 14 Center for Biodefense and Emerging Infectious Diseases, University of Texas Medical Branch, Galveston, Texas, United States of America, 15 Department of Pediatrics and Emory Vaccine Center, Emory University, Atlanta, Georgia, United States of America

☯ These authors contributed equally to this work.
¤a Current address: Molecular Genetics and Microbiology Graduate Program, Duke University, Durham, North Carolina, United States of America
¤b Current address: Department of Microbiology and Immunology, University of Maryland, College Park, Maryland, United States of America
¤c Current address: Microbiology and Molecular Genetics Graduate Program, Emory University, Atlanta, Georgia, United States of America
* Emily.bruce@med.uvm.edu

## Abstract

The evolution of SARS-CoV-2 variants and their respective phenotypes represents an important set of tools to understand basic coronavirus biology as well as the public health implications of individual mutations in variants of concern. While mutations outside of spike are not well studied, the entire viral genome is undergoing evolutionary selection, with several variants containing mutations in the central disordered linker region of the nucleocapsid (N) protein. Here, we identify a mutation (G215C),

**Data availability statement:** All raw sequencing data are available in the NCBI Bioproject ID PRJNA1083584. Individual peptides identified in proteomics experiments are included in S2 Data. Raw proteomics data are available in jPOSTrepo (JPST003506). Plasmids will be provided upon request to the corresponding author, after completion of a material transfer agreement. Infectious virus will be provided after completion of a material transfer agreement and confirmation of suitable BSL-3 facilities and IBC approval to safely handle infectious SARS-CoV-2.

**Funding:** This study was supported by an American Heart Association predoctoral fellowship (HWD; 10.58275/AHA.23PRE1020524.pc.gr.161030), George Mason University Fast Grant (PJB), NIH (NIAID) Award AI165075 (PJB), NIH (NIAID) Award 1R01AI153602-01 (VDM), Investigators in the Pathogenesis of Infectious Disease, Burroughs Wellcome Fund (VDM), and NIH (NIGMS) award P20GM125498 (EAB) and UVM start-up funds (EAB). The protein mass spectrometry was supported by an Institutional Development Award (IDeA) from the National Institute of General Medical Sciences of the National Institutes of Health under grant number P20GM103449. The funders played no role in study design, data collection and analysis, decision to publish, or preparation of the manuscript.

**Competing interests:** I have read the journal's policy and the authors of this manuscript have the following competing interests: HCK, HWD, BAJ, VDM, and EAB have filed a patent on the use of mutations in the nucleocapsid linker as a means of increasing nucleocapsid protein levels. VDM has filed a patent on the reverse genetics system and reporter SARS-CoV-2. Other authors declare no competing interests.

**Abbreviations:** ALI, air–liquid interface; CPE, cytopathic effect; DMEM, Dulbecco's Modified Eagle Medium; DMV, double membrane vesicle; FBS, fetal bovine serum; FFU, focus forming unit; HBECs, human bronchial epithelial cells; LoD, limit of detection; N, nucleocapsid; NEM, N-ethylmaleimide; NGS, Next Generation Sequencing; PCR, polymerase-chain reaction; PEG, polyethylene glycol; TEM, transmission electron microscopy; UTMB, University of Texas Medical Branch; VLP, virus-like particle.

characteristic of the Delta variant, that introduces a novel cysteine into this linker domain, which results in the formation of a more stable N-N dimer. Using reverse genetics, we determined that this cysteine residue is necessary and sufficient for stable dimer formation in a WA1 SARS-CoV-2 background, where it results in significantly increased viral growth both in vitro and in vivo. Mechanistically, we show that the N:G215C mutant has more encapsidation as measured by increased RNA binding to N, N incorporation into virions, and electron microscopy showing that individual virions are larger, with elongated morphologies.

## Introduction

The coronavirus disease of 2019 (COVID-19) pandemic originated from the emergence of Severe Acute Respiratory Syndrome Coronavirus 2 (SARS-CoV-2) in late 2019 [1]. Subsequent worldwide spread and sustained transmission over the past 4 years, combined with near real-time access to viral genomic surveillance data, has revealed a detailed picture of SARS-CoV-2 evolution in the human population. Individual mutations that conferred an evolutionary advantage were quickly selected and maintained in viral lineages, leading to the emergence of novel Variants of Concern distinguished by increased immune evasion, transmission, disease burden, or infectivity. Understanding the biological function of the individual mutations that led to the emergence of novel variants of concern has important implications for public health consequences as well as our fundamental understanding of basic coronavirus biology.

Most studies concerning these mutations have focused on genetic changes within the spike (S) protein, due to concerns that sufficiently novel spike proteins can allow viral escape from the immune memory induced by vaccination or prior infection [2–5]. However, mutations elsewhere in the viral genome can play key roles in viral replication and pathogenesis [6–8]. The nucleocapsid (N) protein in particular is a genetic "hotspot" for mutations across variants, with several mutations within the flexible linker region associated with increased replication and pathogenesis [8–11]. The Alpha, Gamma, and Omicron variants contained the R203K/G204R mutations, while the Delta (G215C) and Lambda (G214C) variants were characterized by mutations inserting cysteines, all into the flexible linker region (summarized in Nygun and colleagues [12], Kahetran and colleagues [13]). Given the many attributed roles of N, including RNA encapsidation [14], production of viral RNA [15–18], and viral assembly through interactions with M [19–21], mutations within this protein are poised to have large impacts on the viral lifecycle.

The Delta variant, which emerged in mid-2021, possessed the ability to evade prior immunity, increased transmission and altered virulence, and contained a series of novel mutations, including key changes in the spike protein [7]. While the function of the mutations in spike has been intensively studied [4,22,23], the majority of Delta sub-lineages also contained a unique mutation in the nucleocapsid protein which converted a glycine into a cysteine at position 215 (N:G215C) [10,24].

Bioinformatic analysis examining the rate of amino acid substitutions among 2.49 million SARS-CoV-2 sequences deposited in GISAID after the introduction of the Delta variant demonstrated that while most amino acid substitutions are transient, the N:G215C mutation was maintained over time in GISAID sequence data, independent of concurrent changes in spike or N [10].

This introduction of a cysteine residue within the nucleocapsid is unusual among pandemic-causing betacoronaviruses and has major structural implications as it could result in the production of a stable N-N dimer linked by a disulfide bond. Here, we describe the impact of this novel cysteine within the N protein and the role it plays in viral replication and particle formation. We demonstrate that this cysteine promoted N oligomerization by stabilizing a covalent N-N interaction that resolves as a dimer on standard non-reducing SDS-PAGE gels. While higher order structures composed of N multimers that disappear upon reduction have been previously reported for both mouse hepatitis virus [25], Bovine Coronavirus, and OC43 [26], in each case, the higher order multimer appears to run as a trimer of N, rather a dimer as observed here. Furthermore, none of these viruses contain a cysteine in the linker region, suggesting that the N-multimerization previously observed for other coronaviruses may be fundamentally different the process driving SARS-CoV-2 nucleocapsid oligomerization.

Several studies published during the time this work were in preparation and review have also elegantly examined the role of the G215C mutation in nucleocapsid function. Early characterization of purified Delta nucleocapsid protein revealed that it was more compact and less discorded than the ancestral WA1 protein, and that the canonical Delta mutation G215C increased N self-association, shifting the protein from a primarily dimeric to tetrameric state via disulfide cross links of noncovalent dimers while increasing its association with nucleic acids [10]. Subsequent studies from the Schuck group identified a highly conserved transient helix structure in a leucine-region of the N linker that can cooperatively assemble into coiled-coil trimers, tetramers, and higher order oligomers in a process that is further enhanced by binding to nucleic acid [27]. The N:G215C mutation stabilizes this helical state (and thus enhanced self-association), which in turn enhanced activity in a virus-like particle (VLP) assay measuring relative expression of a reporter gene packaged into VLPs created by transfecting the four main structural proteins into HEK 293T cells [28].

Our work is the first to examine the role of the N:G215C mutation in the context of fully replicating virus and demonstrates that the N:G215C mutation that was a defining mutation of the Delta variant resulted in substantially increased viral replication kinetics in primary differentiated human bronchial cells. The N:G215C mutation also increased viral replication in the nasal washes and lungs of infected Syrian golden hamsters while paradoxically delaying weight loss. Finally, the N:G215C virus increased the association between N and RNA in virions, packaged substantially more N per virion and resulted in many of the virions displaying an elongated morphology. Altogether, our data suggest that the Delta N:G215C mutation increases levels of nucleocapsid oligomerization which drives increased packaging of N into mature virions and results in significant increases in viral replication both in vitro and in vivo.

## Results

### Introduction of a novel cysteine in the nucleocapsid linker region of variants of concern

The G215C mutation in nucleocapsid is a defining feature of the Delta variant lineages (Fig 1A) and sits in the disordered linker region of the nucleocapsid protein, which lies between a N-terminal RNA-binding domain and a C-terminal dimerization domain (Fig 1B). The introduction of a cysteine in the SARS-CoV-2 nucleocapsid is unique amongst zoonotic Betacoronavirus, as neither SARS-CoV, MERS, nor other SARS-CoV-2 variants' nucleocapsids contained any cysteines (Fig 1C). Furthermore, while other coronavirus nucleocapsid proteins do contain cysteines, they are largely absent from the linker region (S1 Fig). Surprisingly, when analyzing the sequences of our panel of VOCs obtained from clinical specimens, we discovered that two other variant (Beta and Iota) isolates also contained a cysteine within the N protein (Fig 1C). Interestingly, all three of these mutations sit within the intrinsically disordered linker region of N (also termed N3/sN3) between N2 (RNA binding domain) and N4 (dimerization domain). Both our Beta (B.1.351) and Iota (B.1.526) stocks

A

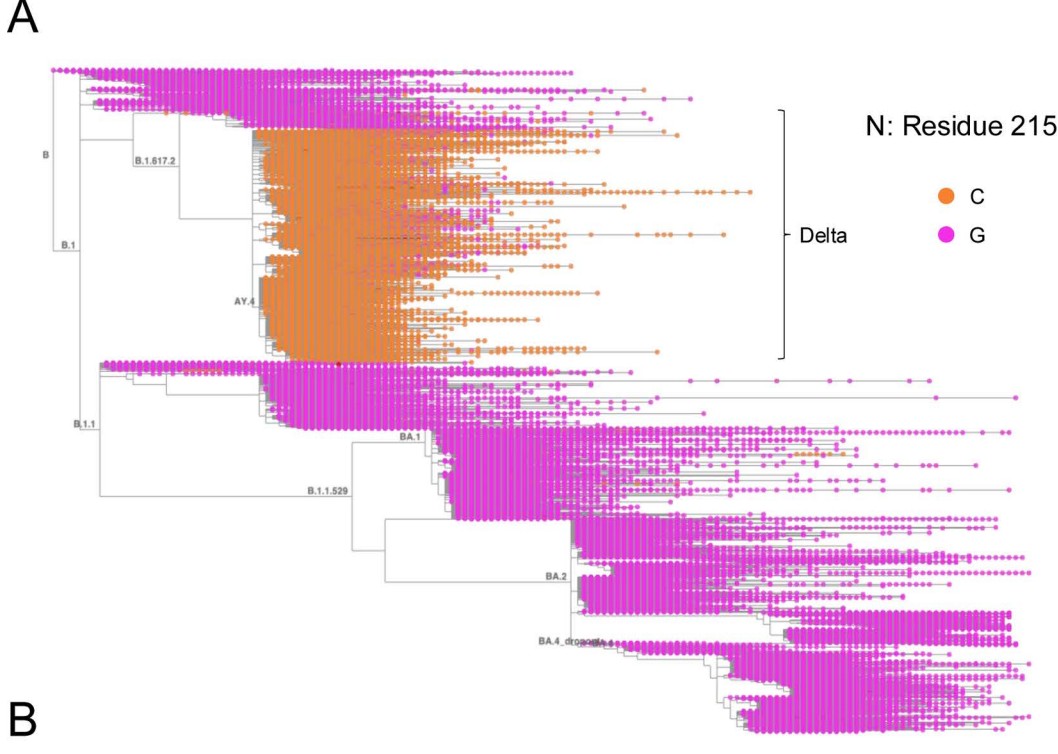

B

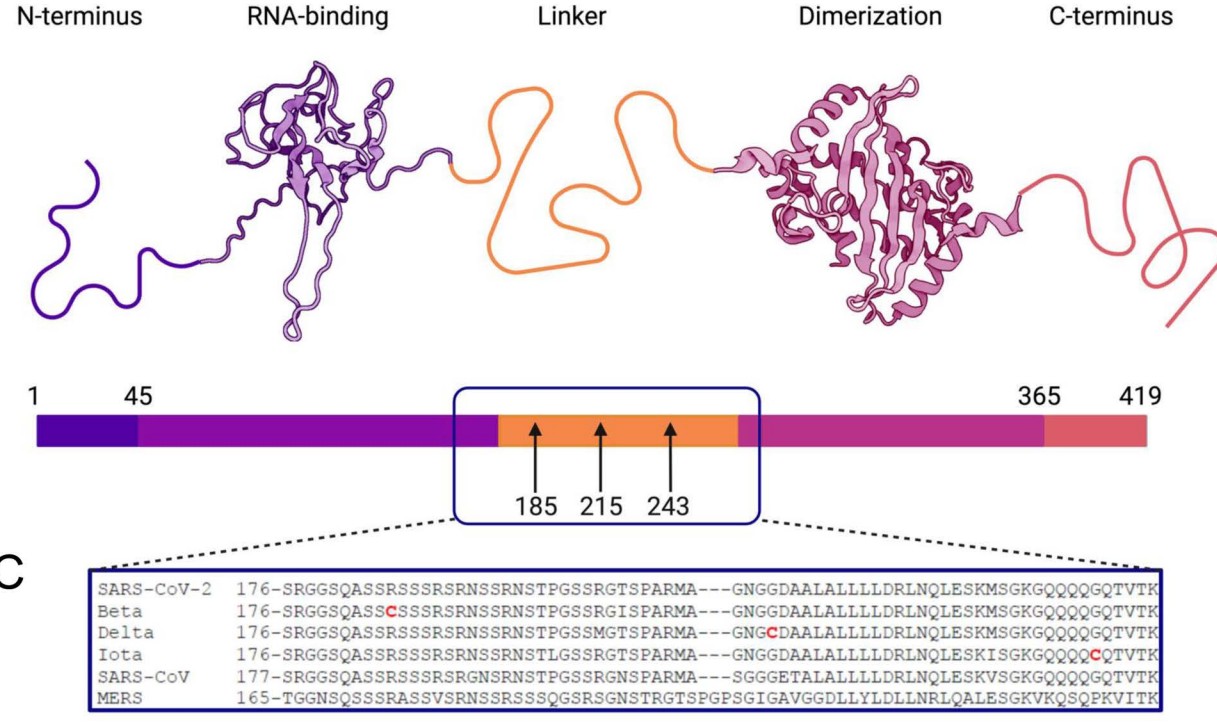

C

```
          176-SRGGSQASSRSSSRSRNSSRNSTPGSSRGTSPARMA---GNGGDAALALLLLDRLNQLESKMSGKGQQQQGQTVTK
SARS-CoV-2
Beta      176-SRGGSQASSCSSSRSRNSSRNSTPGSSRGISPARMA---GNGGDAALALLLLDRLNQLESKMSGKGQQQQGQTVTK
Delta     176-SRGGSQASSRSSSRSRNSSRNSTPGSSRGTSPARMA---GNGCDAALALLLLDRLNQLESKMSGKGQQQQGQTVTK
Iota      176-SRGGSQASSRSSSRSRNSSRNSTLGSSRGTSPARMA---GNGGDAALALLLLDRLNQLESKISGKGQQQQCQTVTK
SARS-CoV  177-SRGGSQASSRSSSRSRGNSRNSTPGSSRGNSPARMA---SGGGETALALLLLDRLNQLESKVSGKGQQQQGQTVTK
MERS      165-TGGNSQSSSRASSVSRNSSRSSSQGSRSGNSTRGTSPGPSGIGAVGGDLLYLDLLNRLQALESGKVKQSQPKVITK
```

**Fig 1. Introduction of novel cysteine residues into the SARS-Co-2 nucleocapsid linker. (A)** Amino acid identity of position 215 in the SARS-CoV-2 nucleocapsid protein in genomic surveillance data from 7,342,041 sequences in The International Nucleotide Sequence Database Collaboration is

shown (Glycine in pink, Cysteine in orange). Sequences were visualized by Taxonium11 on January 10th, 2024, at position 215 of the N protein. The tree was rooted to Wuhan/Hu-1 (GenBank MN908947.3, RefSeq NC_045512.2), sequences were added to the tree through the use of UShER12, and Pangolin lineage identity is labeled (B.1617.2 branch shows Delta sequences and B.1.1.529 shows Omicron sequences). **(B)** The SARS-CoV-2 N protein includes an unstructured N-terminus (residues 1−45; in indigo), an N-terminal RNA binding domain (NTD; residues 45−176; in violet, PDB code 6YI3 [80]), an unstructured linker (LKR) region (176−263, shown in yellow) which binds to Nsp3, and a C-terminal dimerization domain (CTD; residues 263−365; in pink, PDB code: 6WZO [9]), followed by an unstructured C-terminal region (residues 265−419; in peach). **(C)** The sequences of the Nucleocapsid linker region from SARS-CoV-1 (WA1), the Beta, Delta, and Iota isolates used in this paper, as well as SARS-CoV and MERS were aligned using Clustal Omega. Cysteines are highlighted in red, and arrows highlight their position in the schematic. Figure created with Biorender.

contained novel cysteines in the linker region, R185C (99.7% of reads) and G234C (100% of reads), respectively (Fig 1C). Since the introduction of a cysteine residue would allow for the formation of a new disulfide-bonded N-N dimer complex, we predicted that this mutation could have major impacts on the secondary, tertiary, and/or quaternary protein structure of the nucleocapsid protein.

### Cysteine in the N linker promotes the formation of a stable N-N dimer

To test if these novel cysteines would make a more stable N-N dimer, we visualized the N from our panel of variant isolates by western blot under non-reducing conditions. We infected VeroE6-TMPRSS2 cells with wildtype virus (ancestral SARS-CoV-2 from the WA1 infectious clone) as well as low passage stocks of the Alpha, Beta, Gamma, Delta, Epsilon, Iota, Mu, and Omicron variants isolated from clinical samples. All viruses produced a band of the expected molecular weight (~47 kDa) for the N monomer, and the majority also showed a series of truncation products (indicated with <) that we hypothesize to be caspase cleavage products [29] (Fig 2A). As predicted, the three variants which contained the novel cysteine residue (Beta, Delta, Iota), produced a second band detected at twice the molecular weight of the expected N monomer (Fig 2A, see bands labeled with the asterisk symbol [*]). Of the three variants, Delta (G215C) produced the greatest level of dimerized-N, with Iota (G243C) and Beta (R185C) each producing slightly lesser amounts (Fig 2B). While it is known that coronavirus N proteins typically form dimers via their dimerization domains, this is mediated by non-covalent bonds and the canonical N-dimer was not seen by SDS-PAGE/western blot for viruses that lacked cysteines (WT, Alpha, Gamma, Epsilon, Mu, and Omicron).

To further test whether this higher molecular weight band represented a disulfide-bonded form of N-N dimer, we performed several additional experiments. First, we prepared samples under strong reducing conditions (10-mM DTT), where we observed the higher molecular weight band was eliminated (S2A Fig). To confirm the putative dimer was not an artifact of the monoclonal antibody used, we probed lysates with a second independent antibody (S2B and S2C Fig). Slight differences in cleavage products were seen with the two antibodies (see *), confirming that the antibodies recognized different epitopes, but the higher molecular weight band was visible in both conditions (S2B–S2D Fig). To ensure that the higher molecular band did not represent a disulfide bond formed between N and a cellular or viral protein of equivalent weight, we immunoprecipitated N from cells infected with WT or Delta SARS-CoV-2 and performed mass spectrometry on tryptic peptides from gel slices cut from the regions corresponding to the monomer and putative "dimer" (S2E Fig). In the Delta infected sample, most peptides detected in the "dimer" gel slice corresponded to N, and roughly half the total N peptides were detected in the "dimer" versus "monomer" slice (S2F Fig). Furthermore, there were no cellular peptides of similar abundance found in the "dimer" slice, with the most abundant cellular protein found at >1/5th the levels of N (S2 Data).

To ensure that this disulfide bond truly occurred within infected cells and was not a post-lysis artifact, lysates were harvested in the presence of increasing concentrations of N-ethylmaleimide (NEM), which binds irreversibly to free cysteines and prevents the formation of post-lysis disulfide bonds. Cells infected with the N:G215C virus, when lysed in our standard triton lysis buffer (Fig 2C, lane 5), showed stably dimerized N. When lysed in increasing concentrations of NEM (Fig 2C; 1–10 mM), the stably dimerized form was still present at similar levels to cells lysed without NEM (lanes 7, 8, versus 6, see quantification in S3 Fig). To further investigate whether the disulfide bonded dimer was present inside infected

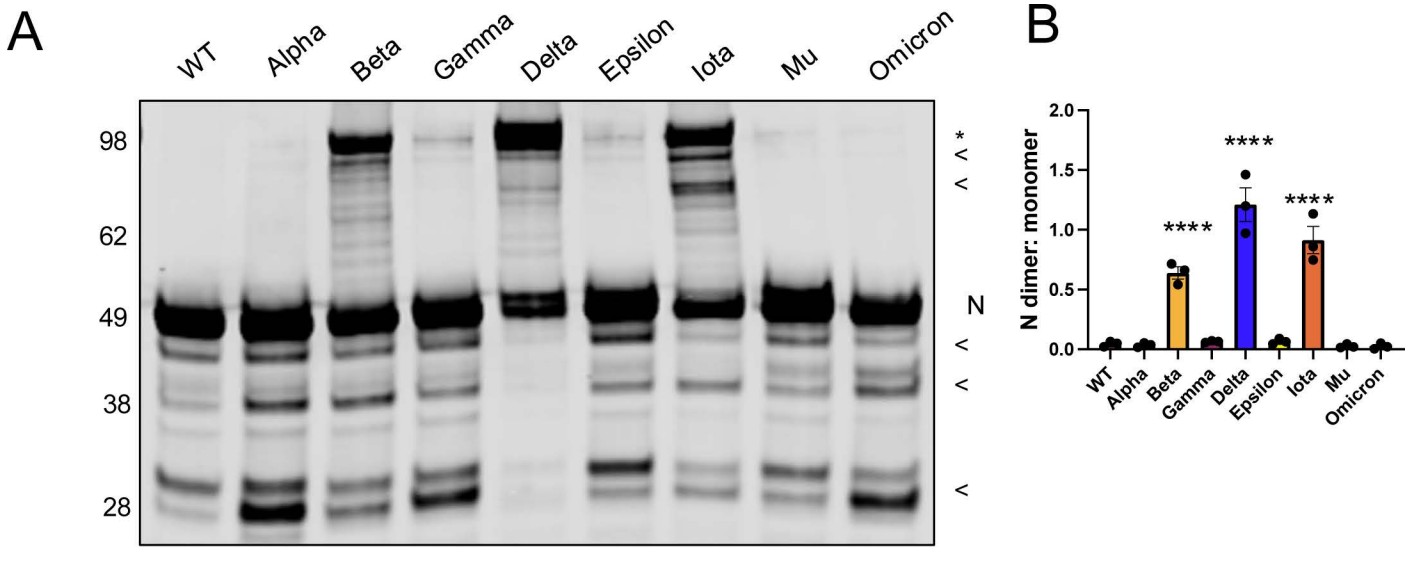

**Fig 2. A disulfide-bonded nucleocapsid dimer is formed in several Variants of Concern.** VeroE6-TMPRSS2 cells were infected with the indicated SARS-CoV-2 variants (or WA1, termed Wild type) at an MOI of 0.01 for 24 h. **(A)** Unreduced cell lysates were visualized by SDS-PAGE and western

blot using antibodies recognizing SARS-CoV-2 N. The presence of a ~100 kDa band recognized by the SARS-CoV-2 N antibody in the Beta, Delta, and Iota variants is indicated by an asterisk (*). A representative gel and **(B)** the relative levels of N dimer to monomer from three independent biological replicates are shown. Mean ± SEM is plotted, individual data points for each experiment are superimposed onto bar graphs for each condition. Statistical comparisons were conducted using a one-way ANOVA with Dunnett's multiple comparisons, ****=$p < 0.0001$. **(C)** VeroE6-TMPRSS2 cells were infected with WT or the G215C virus at an MOI of 0.01 for 24 h. Cell lysates were harvested in standard triton lysis buffer (lanes 1, 5) or in the presence of increasing concentrations of N-ethylmaleimide (1, 10, 100 mM). **(D)** A parallel experiment was performed in which cells were pre-treated by incubating for 30 min at 37°C in increasing concentrations (1, 10, 100 mM) of NEM in PBS. Cells were then lysed as in **(C)**, with NEM present in the lysis buffer as well as the pre-incubation. Lysates were visualized by SDS-PAGE and western blot using antibodies to SARS-CoV-2 N and actin. The presence of a ~100 kDa band recognized by the SARS-CoV-2 N antibody in the G215C mutant is indicated by an asterisk (*). A representative western blot from two independent biological experiments is shown. The data underlying this Figure can be found in S1 Data.

cells *before* lysis, we incubated cells with increasing concentrations of NEM before lysing cells in the same lysis buffer as above, including fresh NEM at the time of lysis (Figs 2D and S3). While a ~80% reduction in dimerized N was observed, stable dimerized N was seen at every NEM dose including a 10-fold excess of typical NEM concentrations. These results indicate covalent bonds contribute to N dimerization of the N:G215C mutant. While some of the linkages in the N dimer band could form post lysis, off target effects of NEM treatment and N dimer formation in late stage infection/assembly could also explain some of the reduction in pre-lysis treated samples. Similarly, while protein oligomers could be resistant to NEM similar to SDS-resistant oligomers [30,31], sensitivity to DTT (Fig 2A) argues against this possibility. In total, our data are consistent with both covalent and non-covalent interactions playing a role in the N-N interaction in the context of viral infection, driven by the cysteine residue at position 215.

### N dimer stability dependent on novel cysteines in linker region

As the production of a disulfide bond in the reducing environment of the cytoplasm is unusual, we next determined whether the formation of this stable N-N dimer required the context of infection. SARS-CoV-2 replication produces many membranous compartments with limited cytoplasmic access that could shield N during authentic infection. We created plasmid expression constructs of the WT, Delta, Beta, and Iota nucleocapsid sequences and transfected these into HEK-293T cells, in conjunction with constructs where each cysteine was mutated back to the canonical residue in the WT sequence (Delta C215G, Beta C185R, Iota C243G). While all three variant constructs (Delta, Beta, Iota) produced stably dimerized N (Fig 3A), Delta maintained the highest levels, while the Beta construct did not consistently form visible, stable dimer (Fig 3B). In each case, when the cysteine was reverted to its original amino acid, the stabilized dimer was not made (Fig 3A and 3B). These data suggest that for Delta and Iota, the presence of other viral proteins/RNA and the formation of double membrane vesicles (DMVs) are not required for stably dimerized N formation. As expressing N protein (in uninfected cells, which lack any additional viral component) results in the formation of a higher molecular weight band, we conclude that the presence of a cysteine at 215/243 in the Delta/Iota backgrounds is sufficient to confer the stable N dimer phenotype.

### Impact of the N:G215C mutation on viral growth in vitro and in vivo

As bioinformatic evidence suggests that Delta variants containing the N:G215C mutation outcompeted those that contained the ancestral glycine at that position [10], we wanted to examine the impact of the N:G215C mutation on viral growth kinetics. Since WT SARS-CoV-2 (WA1/2020) does not produce the stably dimerized N in infection (Fig 2A), we investigated if the introduction of a cysteine at 185, 215, or 243 (Beta, Delta, Iota) was sufficient to mediate stable dimer formation in a WT background. Using plasmid constructs, we transfected HEK-293T cells with either WT N or WT N containing a single point mutation (R185C, G215C, and G234C). When visualized via western blot in non-reducing conditions, the construct which contained the Delta G215C mutation most robustly produced the stabilized dimer across biological replicates, suggesting that this mutation is both necessary and sufficient for stable dimer formation (Fig 3C and 3D).

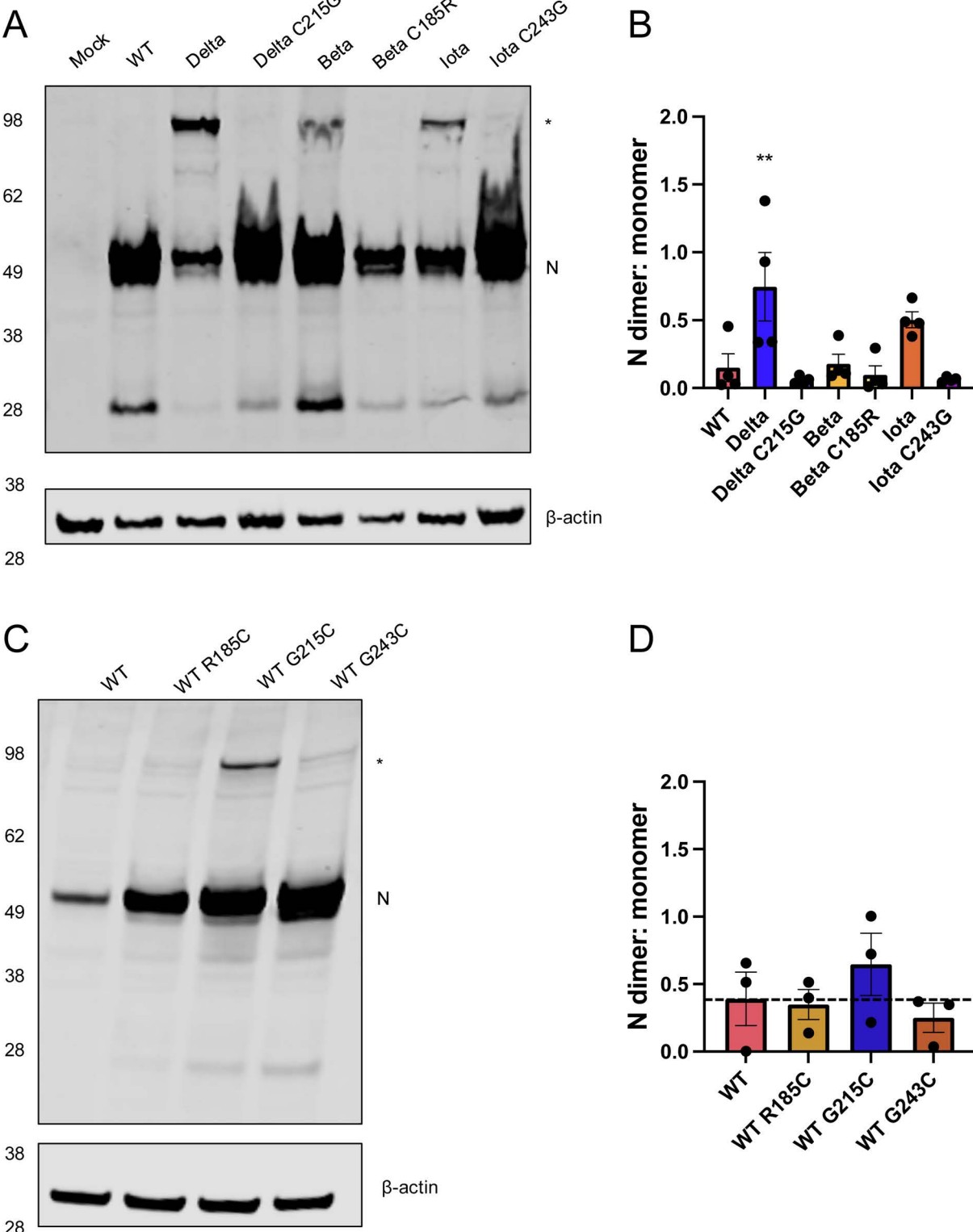

**Fig 3. N-linker cysteine residues are necessary and sufficient for stable dimer formation. (A)** HEK-293T cells were transfected with plasmids encoding the N protein of the indicated SARS-CoV-2 variant. In addition, cells were transfected with constructs in which each cysteine was changed

back to the parental residue in the WT (WA1) sequence. Twenty-four hours post transfection, unreduced cell lysates were visualized by SDS-PAGE and western blot using antibodies recognizing SARS-CoV-2 N and β-actin. **(B)** The ratio of N monomer to dimer bands seen in the western blot shown in **(A)** was quantified. **(C)** HEK-293T cells were transfected with plasmids encoding the WT N protein or constructs introducing a cysteine at positions 185, 215, or 243, and **(D)** the ratio of monomer to dimer bands was quantified. The presence of a ~100 kDa band recognized by the SARS-CoV-2 N antibody corresponding to the stable dimer is indicated by a *. A representative western from four (A/B) or three (C/D) independent biological experiments is shown, probed for N (red) and β-actin (green). Mean ± SEM is plotted and individual data points for each experiment are superimposed onto bar graphs for each condition. Statistical comparisons were conducted using a one-way ANOVA with Dunnett's multiple comparisons, **=0.0056. The data underlying this Figure can be found in S1 Data.

As inserting the G215C mutation in the WT background was sufficient to confer dimer formation, we used a well-established reverse genetics system [32] to rescue an infectious clone containing the G215C nucleocapsid mutation (N: G215C) in the SARS-CoV-2 WA1 backbone (Fig 4A). We used a neon-green reporter virus in the WA1 background (mNG SARS-CoV-2), which is genetically identical to WA1 except that ORF7 has been replaced with a neon green fluorescent reporter. Notably, mNG SARS-CoV-2 is attenuated compared with the parental WA1, due to the replacement of ORF7a with the mNG reporter [8]. Due to their permissiveness and ability to support high levels of viral replication, we first characterized the activity of the newly rescued virus in Vero-TMPRSS2 cells. As expected, the N:G215C virus produced the stable N dimer in infected Vero-TMPRSS2 cells under non-reducing conditions (Fig 4B). In a multi-cycle growth curve, the WT and N:G215C viruses grew to identical peak titers, with very similar growth kinetics. We did note that at the earliest time point (7hpi), the WT virus shows an ~1 log drop in viral titer (representing the loss in infectivity of the original inoculum), while the N: G215C virus did not display such a drop (Fig 4C), though the biological significance (if any) of this observation is unclear.

Next, as VeroE6 cells are African green monkey kidney cells that lack a functional type I IFN system, we examined viral growth kinetics in primary differentiated human bronchial epithelial cells (HBECs), grown in transwells on an air–liquid interface (ALI). As expected, the N:G215C virus produced the stable N dimer in infected HBECs when harvested under non-reducing conditions (Fig 4D). Notably, in multi-cycle growth curves, the mNG N:G215C virus had improved growth kinetics, with a peak titer more than 100 times greater than the mNG WT virus (Fig 4E). These data suggest that the stably dimerized form of N conveys a particular advantage to viral replication in primary differentiated human bronchial cells.

Given the strength of this phenotype in primary human cells, we next determined the effect of the N:G215C mutation in vivo in the Syrian Golden Hamster model of SARS-CoV-2 infection. Three- to four-wk-old male hamsters were inoculated intranasally with either PBS (mock), $10^4$ PFU of the WT neon green reporter SARS-CoV-2 (mNG WT), or $10^4$ PFU of the neon green reporter SARS-CoV-2 containing the N:G215C mutation (N:G215C). Animals were monitored for weight gain/loss daily for 7 days, and cohorts of five animals underwent nasal washing followed by euthanasia to obtain tissues to determine viral loads in the lung at both day 2 and day 4 post infection (Fig 5A). On day 7, surviving animals were euthanized and lung tissue collected for virological and histopathological analysis. While animals infected with the N:G215C mutation showed significant weight loss, relative to control WT virus, the kinetics were delayed with peak disease achieved at days 5–6 post infection, 1–2 days after WT infection (Fig 5B). Strikingly, despite delayed weight loss, viral replication was increased with the G215C mutation. Modest but significant increases in viral titers were observed in the nasal washes at day 4, and a sustained 10-fold increase over WT titers in the lungs throughout the infection (Fig 5C and 5D). Together, despite the kinetic delay, the N:G215C mutant caused similar overall weight loss and augmented viral replication.

Examining histopathology, the N:G215C mutant had modest changes in antigen staining but increased infiltration and damage relative to WT control virus. At day 2, both WT and N:G215C had similar antigen distribution and scores (S5A and S5B Fig). By day 4, N:G215 had a modest increase in overall antigen staining mostly driven by significant differences in airway staining. Viral staining was cleared in both WT and mutant infected animals by day 7 post infection. Examining immune infiltration and damage, lesions were of similar composition and size at day 2 for both groups, but more severe in WT animals

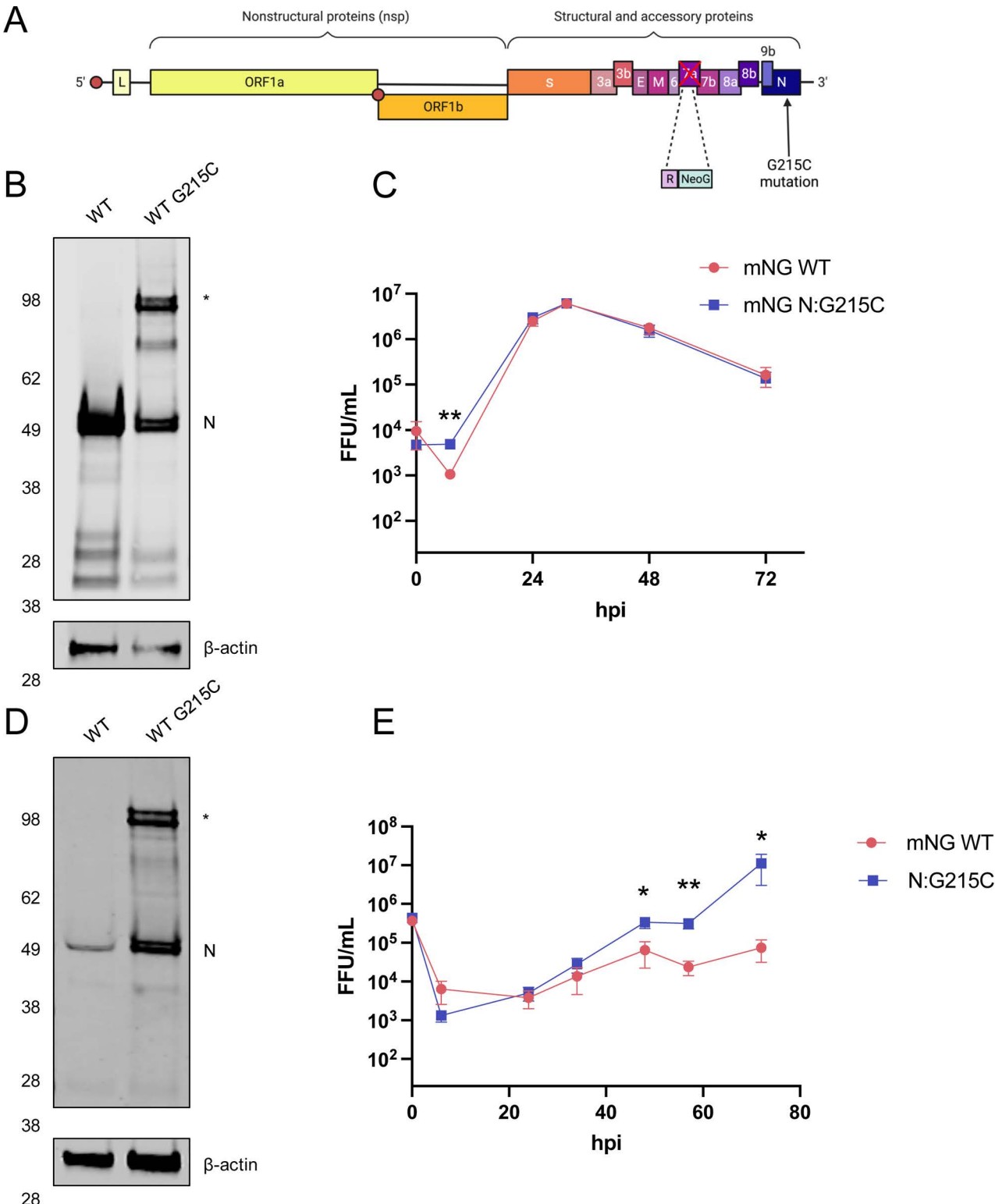

**Fig 4. The introduction of N:G215C in a WA1 infectious clone recapitulates stable N dimer formation and displays altered growth kinetics in HBECs. (A)** A schematic of the SARS-CoV-2 genome encoding the mNG infectious clone system is shown, including the replacement of ORF7a with mNeon Green, and the introduction of the G215C mutation in N. **(B)** VeroE6-TMPRSS2 cells were infected, or mock infected, with the WT or N:G215C

viruses at an MOI of 0.0005 for 1 h. Seventy-two hpi, unreduced cell lysates were collected and visualized by SDS-PAGE and western blot using a rabbit anti-SARS-CoV-2 N and β-actin. The presence of the higher MW (~100 kDa) band seen in the N:G215C virus is indicated with a *. A representative gel from three independent biological experiments is shown. **(C)** In addition, viral supernatants were collected at 8, 24, 32, 48, and 72 h post infection and titered by focus forming assay (FFU; focus forming unit). **(D)** Human bronchial epithelial cells were infected with WT or N:G215C viruses at an MOI of 0.5 for 1 hour. D) Unreduced cell lysates were collected and processed for western blot as in **(B)**, while **(E)** apical viral washes were collected sequentially from the same well at 8, 24, 32, 48, and 72 h post infection and titered by focus forming assay (FFU). Mean ± SEM is plotted, $N = 6$ from three biological experiments **(C)**, $N = 12$ from four biological experiments **(E)** is shown. Statistical comparisons were conducted using a two-tailed $T$ test for each time point, (* [$p < 0.05$], ** [$p < 0.01$]). Limit of detection (LoD) is 20 FFU/mL and the y-axis minimum is set as the LoD. The data underlying this Figure can be found in S1 Data.

(Fig 5E and 5F). However, at day 4, N:G215C had increased infiltration and damage compared with WT infected animals. While interstitial pneumonia, bronchiolitis, and periarterial edema was common in both groups, N:G215C infected mice were consistently observed to have epithelial cytopathology and subendothelial mononuclear cells. Similarly at day 7, N:G215C maintained evidence of significant damage with continued peribrochiolitis and epithelial cytopathology; in contrast, WT infected hamsters had reduced overall damage scores. Altogether, the results demonstrate that despite similar weight loss, the N:G215C mutant infected animals have increased viral antigen accumulation and damage in the lung as compared to control.

## Wildtype SARS-CoV-2 containing the G215C mutation packages more nucleocapsid protein per virion and displays oblong morphology

The SARS-CoV-2 nucleocapsid, such as nucleocapsids of other Betacoronaviruses, is a highly multifunctional protein. Like other coronavirus nucleocapsids, it is thought to play key roles in the packaging of viral RNA [33], the production of viral RNA through interactions with the replication-transcription complex [15–18], and the antagonism of the innate immune response [34–36]. To better understand why the N: G215C mutation was important at a molecular level, we looked at where in the viral life cycle the stably dimerized form of N was observed. We first observed the stable N-dimer in lysates of cells infected with the Beta, Delta, and Iota variants (Fig 2A), though cells transfected with the Delta (and to lesser extent Iota) nucleocapsids were able to form the durable N-dimer even in the absence of other viral machinery (Fig 3A and 3B). As we had observed a gradient of dimer formation depending on where the cysteine mutation was located (215>243>185; Figs 2B and 3B), we next explored whether the stable G215C N-dimer was found at highest levels in transfected cells, infected cells, or in concentrated extracellular virions. Due to the difficulty in transfecting primary differentiated bronchial cells and lack of sufficient material for concentration, we performed these experiments in Vero-TMPRSS2 cells. Accordingly, we measured the ratio of N dimer:N monomer visible on a western blot of samples collected under non-reducing conditions from transfected HEK-293T cells, infected Vero-TMPRSS2 cells, or extracellular virus concentrated by polyethylene glycol (PEG) precipitation (Fig 6A–6C). Interestingly, we noted that the ratio of dimer to monomer was particularly high in extracellular virus (Fig 6C and 6D), though infection (versus transfection in isolation) also appeared to promote formation of the stably dimerized N (Fig 6B versus 6A).

As this enrichment of dimerized N in virions suggested a potential role in encapsidation, we next compared the incorporation of total levels of N to levels of M in the WT versus N: G215C viruses (Fig 6D and 6E). Finally, we compared the levels of N to the number of infectious focus forming units (FFUs) for the two viruses (Fig 6F). In both cases, the N:G215C virus appeared to over-incorporate N in virions, compared with another structural protein (M) or infectious units, suggesting that the stably dimerized N has increased encapsidation activity compared with the WT nucleocapsid protein. To better link the observed dimer formation to biological function, we concentrated virus from the WT and G215C viruses, affinity purified nucleocapsid protein from each (S4 Fig), and used RT-qPCR to quantify the amount of viral RNA bound to the respective nucleocapsids (Fig 6G). As expected, we saw a significant (3- to 4-fold) increase in the amount of viral RNA bound to the G215C nucleocapsid. This result shows that the G215C nucleocapsid protein has increased affinity for viral RNA, which supports our hypothesis that the G215C mutation increases encapsidation.

PLOS Biology

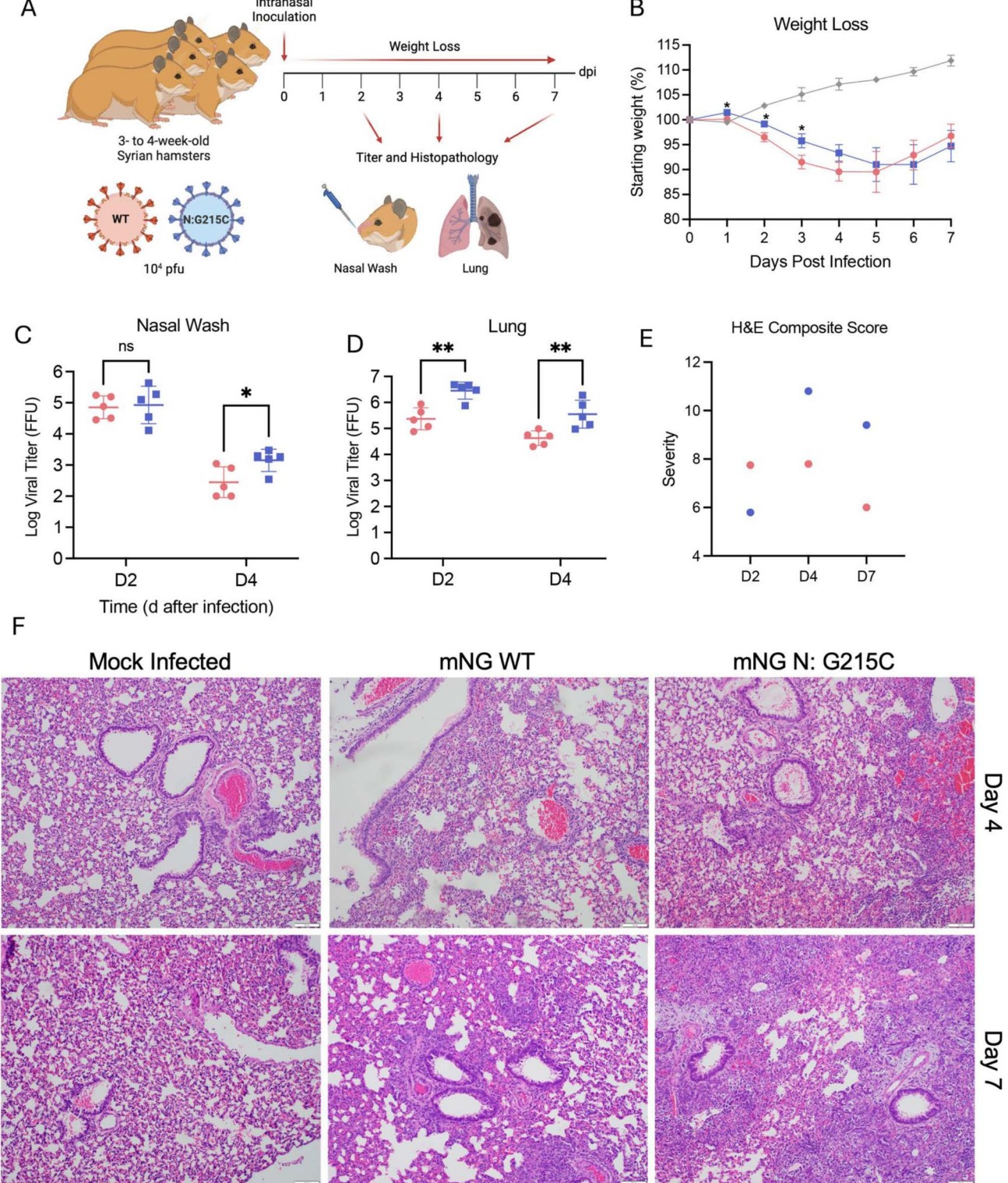

**Fig 5. The N:G215C mutation increases viral growth in nasal washes and the lungs of Syrian golden hamsters as well as immune infiltration and damage. (A)** Three- to four-wk-old male hamsters were intranasally infected with PBS alone (mock—gray) or $10^4$ FFU of WT (pink) or N:G215C (blue) m Neon Green (NG) SARS-CoV-2. **(B)** Weight-loss of infected animals was monitored daily for 7 days post infection. On days 2 and 4 post

infection, titer in the nasal wash **(C)** and right lung **(D)** was determined. On days 4 and 7 post infection, left lung tissue was harvested, fixed, cut into 5-μM section, stained with hemoxylin and eosin, scored for pathological severity **(E)**, and imaged **(F)**. For weights, graphs represent mean weight change ± SEM. For viral titers, lines represent mean viral titer ± SD. Statistical differences were determined by student $T$ test, *$p < 0.05$, **$p < 0.01$. Scale bars are 100 μm. The data underlying this Figure can be found in S1 Data.

Parallel work published while this study was under preparation and review suggested that the G215C mutation enhanced the non-covalent coiled-coil oligomerization of N critical for packaging. These studies proposed that the oligomerization was driven primarily by non-covalent (versus covalent) interactions, with a serine able to functionally substitute for a cysteine at this position in vitro [27,28]. To evaluate the impact of a serine at this position in the context of virus, we generated a SARS-CoV-2 N:G215S mutant and characterized its replication and the packaging of N into virions (S6 Fig). While the N:G215S virus grew with identical titers to WT in Vero-TMPRSS2 cells (S6A Fig), the mutant failed to replicate in primary differentiated human bronchial epithelial (HBE) cells (S6B Fig). Next, we concentrated extracellular virus from Vero-TMPRSS2 cells to examine the levels of N and M seen in virions (S6C Fig). The G215S virions had reduced nucleocapsid compared with both a separate structural protein (S6D and S6M Fig) and infectious units (S6E Fig; PFU). The fact that G215S underpackages N in virions suggests that covalent interactions at the G215C position may allow N oligomerization and encapsidation, even in the context of fully replicating virus. However, the failure of the G215S virus to replicate in primary differentiated HBEs likely reflects an additional role for covalent N-N interactions mediated by a cysteine lost in the context of a serine at that position. Alternatively, the 215S mutation may disrupt kinase and 14-3-3 recognition motifs necessary to balance N phosphorylation and regulate its function [8,37–40]. Importantly, the covalent interaction of the G215C mutation allows for increased N oligomerization and encapsidation without the loss of replication capacity observed in the G215S mutant.

Having observed increased oligomerization and encapsidation with mutation at N position 215 position, we next set out to examine the changes in encapsidation and packaging driven by G215C on a single virion level. We performed thin section transmission electron microscopy (TEM) on Vero-TMPRSS2 cells that were infected with the WT or N:G215C virus. We examined the morphology of mature virions that had completed budding into intracellular compartments and were no longer attached to the cellular membrane, assessing virion shape as well as the amount and arrangement of internal nucleocapsid structures inside individual virions (Fig 7). The WT virions were largely round and ~60 μm in diameter, with electron dense complexes likely representing ribonucleoprotein (N+RNA) complexes packed inside (Fig 7A). The N:G215C virus produced some spherical particles similar to WT virions, but it also produced a substantial fraction of virions that displayed oblong or elongated morphologies, and were larger than the WT virions in circumference (Figs 7A, 7B, and S7). These virions also appeared to package more ribonucleoprotein complex structures than the smaller spherical virions, in agreement with our previous biochemical analysis of bulk virions.

## Discussion

In this study, we investigate the role of unique cysteine residues that were inserted into the SARS-CoV-2 nucleocapsid linker domain in several Variants of Concern. We demonstrate that these cysteines can form stable disulfide bonds that link nucleocapsid proteins together, outside of the canonical dimerization domain. Sequencing data from public health surveillance efforts suggested that the insertion of a cysteine at position 214 (Lambda) and 215 (Delta) was maintained in these two lineages (Fig 1), and our data support the observation that the G215C mutation is beneficial to the virus in vitro (Fig 4E) and in vivo (Fig 5B–5D). While Betacoronaviruses are known to form dimers, this process is generally thought to occur via a dimerization domain in the C terminus of the protein and importantly is not normally the result of a disulfide bond. Due to the location of the two conserved cysteine residues (214 and 215) in the center of the flexible N linker, and prior data suggesting the linker can play a role in oligomerization [9,10], we propose that these mutations help stabilize the association between pairs of N-N dimers and mediate higher order N oligomerization during the assembly phase

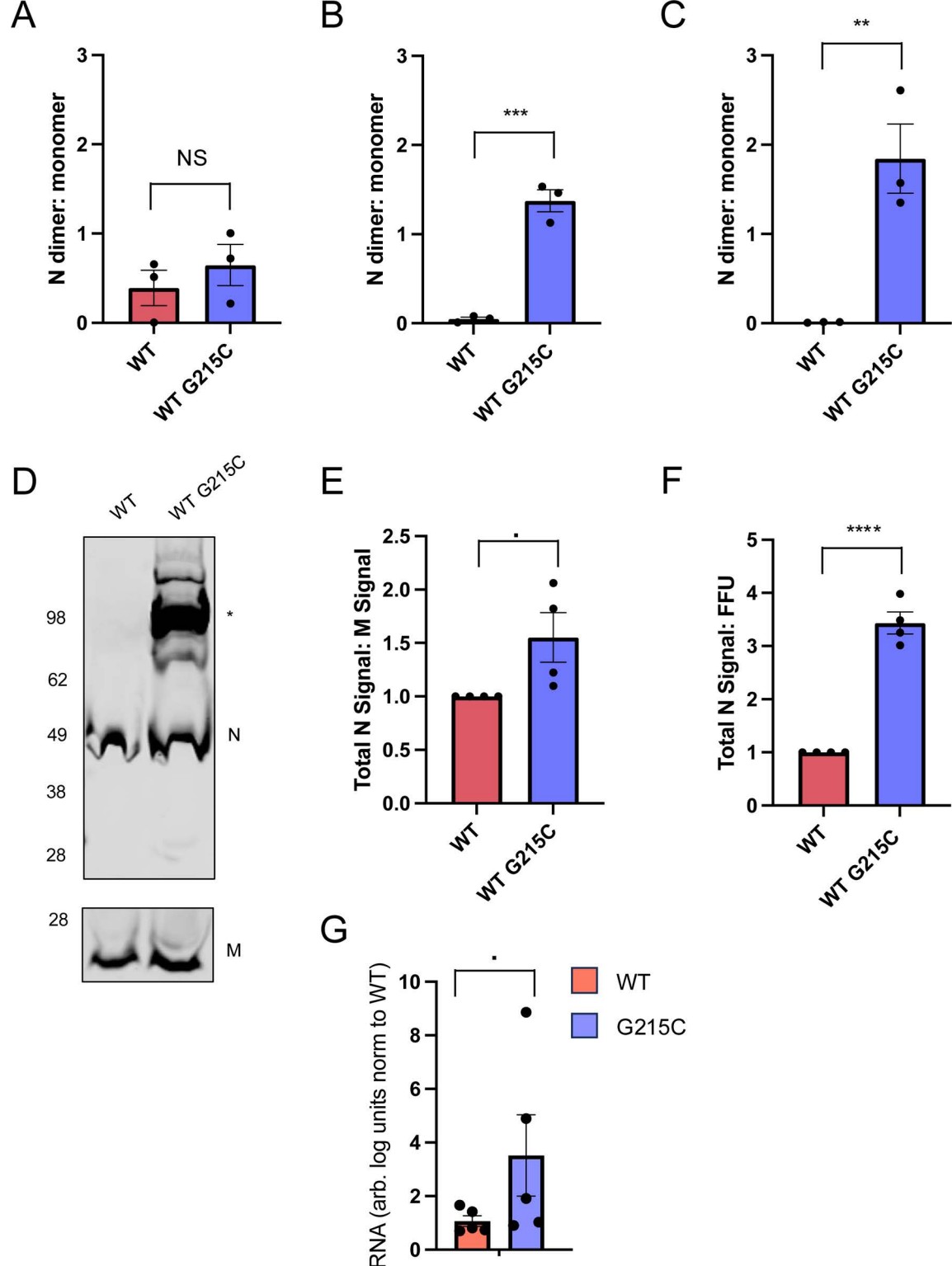

**Fig 6. Stably dimerized N is found preferentially in virions and the G215C mutation results in increased viral packaging of N.** **(A)** HEK-293T cells were transfected with plasmids encoding the WT or G215C N protein. **(B)** Vero-TMPRSS2 cells were infected with the WT or N:G215C viruses at an MOI of 0.0005 for 1 h. **(C)** High titer viral stocks of the WT or N:G215C viruses were concentrated by binding to 10% polyethylene glycol (PEG) then

centrifugated at 10,000*G* for 30 min at 4ºC. Unreduced lysates were collected 24 h post transfection, 72 h post infection or directly from the concentrated viral pellet, visualized by SDS-PAGE, and the ratio of N monomer to dimer was quantified. **(D)** A representative western of **(C)** is shown, probed for N and M. **(E)** The ratio of N–M (normalized to the WT N:M ratio for each replicate) is shown, as well as **(F)** the ratio of N to FFU for each stock (again normalized to the N:FFU ratio in WT virus for each replicate). **(G)** VeroE6-TMPRSS2 cells were infected with WT or the G215C viruses at an MOI of 0.01 for 48 h. Clarified viral supernatants were concentrated by binding to PEG and centrifuging at 10,000*G* for 30 min at 4°C. Concentrated virus was lysed and nucleocapsid proteins from each flask were affinity purified. RNA bound to purified nucleocapsid was extracted and RT-cPCR was used to determine the levels of SARS-CoV-2 RNA bound to N for each condition. $N = 3$ for **A–C**, $N = 4$ for **D–F**, and $N = 5$ for **G**. Mean ± SEM is plotted, individual data points for each experiment are superimposed onto bar graphs for each condition. Statistical comparisons were performed using a two-tailed *T* test, (•[$p = 0.05$-0.1), *[$p < 0.05$], **[$p < 0.01$], ***[$p < 0.001$], or ****[$p < 0.0001$]). The data underlying this Figure can be found in S1 Data.

of the viral life cycle (Fig 8). While our data show that this mutation increases N-N association in both a covalent and non-covalent manner, further work is needed to establish the relative roles of covalent versus non-covalent linkage at this residue. Increasing the affinity of the inter-linker interactions that mediate oligomerization could have the effect of shifting the balance to N toward higher levels of oligomerization. This theory is supported by recent work showing that conserved but transient helices in the leucine-rich sequence of the linker can assemble cooperatively into higher order structures (including trimers, tetramers, and higher oligomers), and stabilizing this structure results in an increase in oligomerization as well as VLP activity [10,27,28]. While these studies were largely conducted on purified protein (either the linker alone, or larger portions of nucleocapsid), our results here suggest that this process is occurring in *bona fide* viral infection, when N will need to successfully carry out a series of different functions to drive a complete cycle of viral replication.

N is a highly multifunctional protein, and it is likely that some of the regulation of its multiple activities depends on whether it is in the monomeric or oligomerized form. It is thought that phosphorylation of N, specifically within the SR region of the linker, acts as a biological "switch" and mediates the switch between genome packaging/assembly and N's intracellular roles (including binding NSP3) [8,41–43]. The G215C Delta mutation lies only slightly upstream of key mutations at 203/204 seen in multiple VOCs that modulate viral pathogenicity [8]. It is possible that oligomerization status and phosphorylation states play interacting roles, and that steric considerations affect the ability of cellular kinases to access these residues in the oligomerized state. Nucleocapsid in mature virions is known to be hypo-phosphorylated in the SR region that encompasses/is adjacent to both the 203/204 and 215 mutations, in contrast to the intracellular pool which is hyper-phosphorylated [44,45]. Our data indicate that the stably dimerized form of N is packaged at higher levels into mature virions, this effect may be partially due to a decreased ability of cellular kinases to access the key phosphorylation sites. It is also possible that the stably dimerized form of N resulting from the G215C mutation slows the rate of uncoating during viral entry, both by increasing affinity of N for RNA and by preventing hyperphosphorylation by the cellular kinases SRPK1/GSK-3/CK-1 which drive a conformational switch in N that abolishes interaction with RNA [46].

One of the key roles of the nucleocapsid is to encapsidate the viral RNA during the process of packaging the full-length viral genome into newly forming virions, helping to chaperone and protect the viral genome in its transition from the producer to new target cell. It is known that the nucleocapsid bound to the viral genome is generally in higher order structures (~14–20 nm, likely composed of 12 copies of N), around which the viral RNA is wrapped (similar to beads on a string) [47–49]. Our data support this framework, as the stably dimerized form of N is seen preferentially in free virus (Fig 6C) compared with transfected or infected cells (Fig 6A and 6B). Increasing the stability of nucleocapsid oligomers could increase the rate or efficiency of viral encapsidation, which fits with our observation that the G215C mutation increases the amount of nucleocapsid packaged into virions, compared with infectious viral units or other viral structural proteins (Fig 6E and 6F) as well as increasing the association with RNA (Fig 6G).

While SARS-CoV-2 virions lack the rigid organization of a virus with a defined icosahedral capsid [50], a 50%–300% increase of nucleocapsid would seem difficult to accommodate within the standard shape a WT virion. Indeed, our data show that, for at least a portion of particles, shifting the oligomerization status by introducing the G215C mutation results in particles that are elongated compared with WT virus (Fig 7). This finding is particularly interesting in light of recent

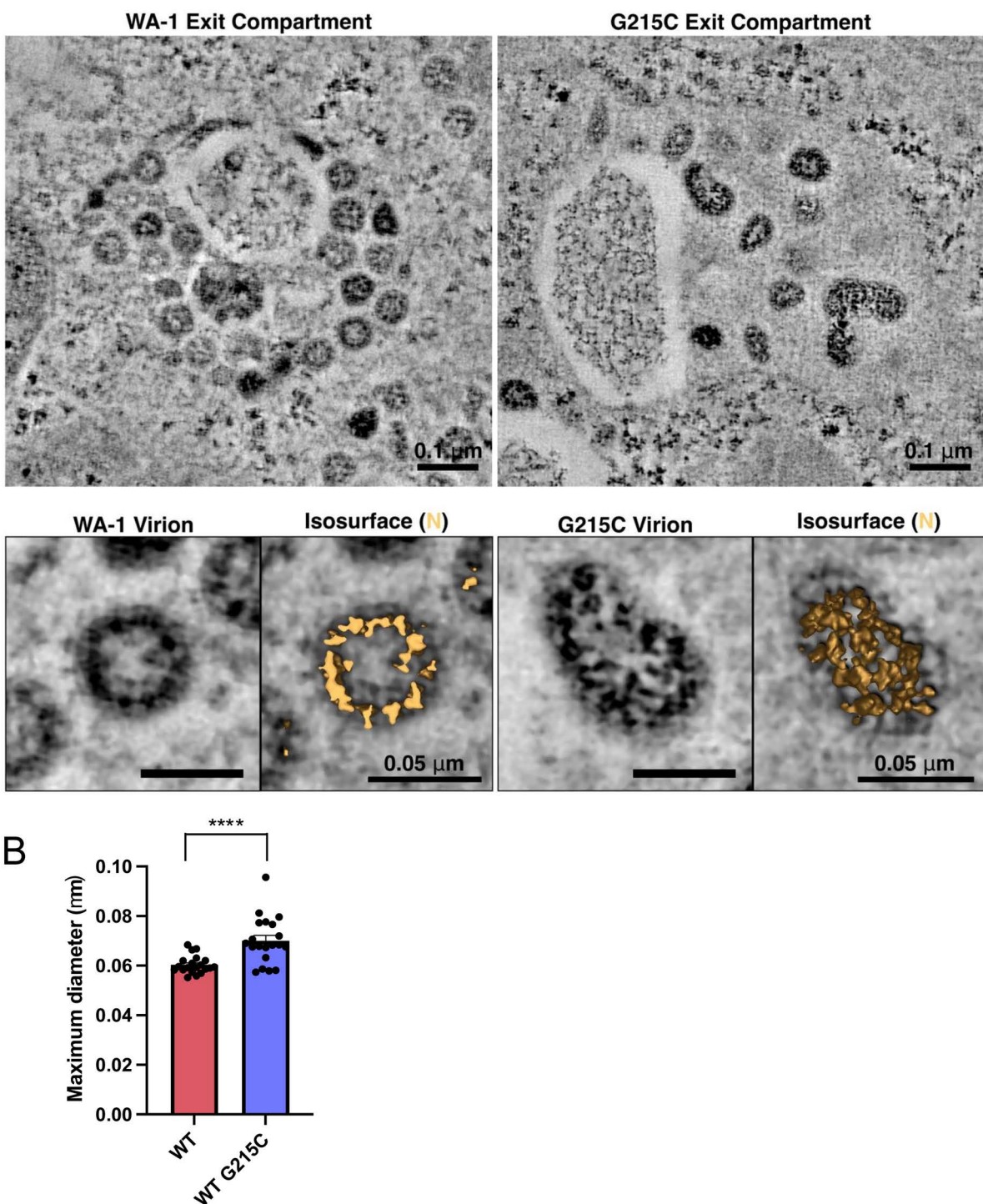

**Fig 7. N:G215C virions are enlarged and show over-incorporation of N. (A)** Vero-TMPRSS2 cells were infected with WT or N:G215C viruses at an MOI of 0.1. The following day cells were prepared for electron microscopy by high-pressure freezing and freeze-substitution, then sectioned and imaged by dual-axis electron tomography. Virus-containing exit compartments were located in both samples, and virions that had completely separated

from cellular membranes were selected and analyzed in 3D in order to determine the structure of intact virions and the arrangement of internal ribonucleoprotein complexes. **(B)** The maximum diameter of 20 randomly selected virions for each virus was measured (see S7 Fig). Mean ± SEM is plotted, individual data points for each experiment are superimposed onto bar graphs for each condition. Statistical comparison was performed using a two-tailed *T* test, **** ($p < 0.0001$). The data underlying this Figure can be found in S1 Data.

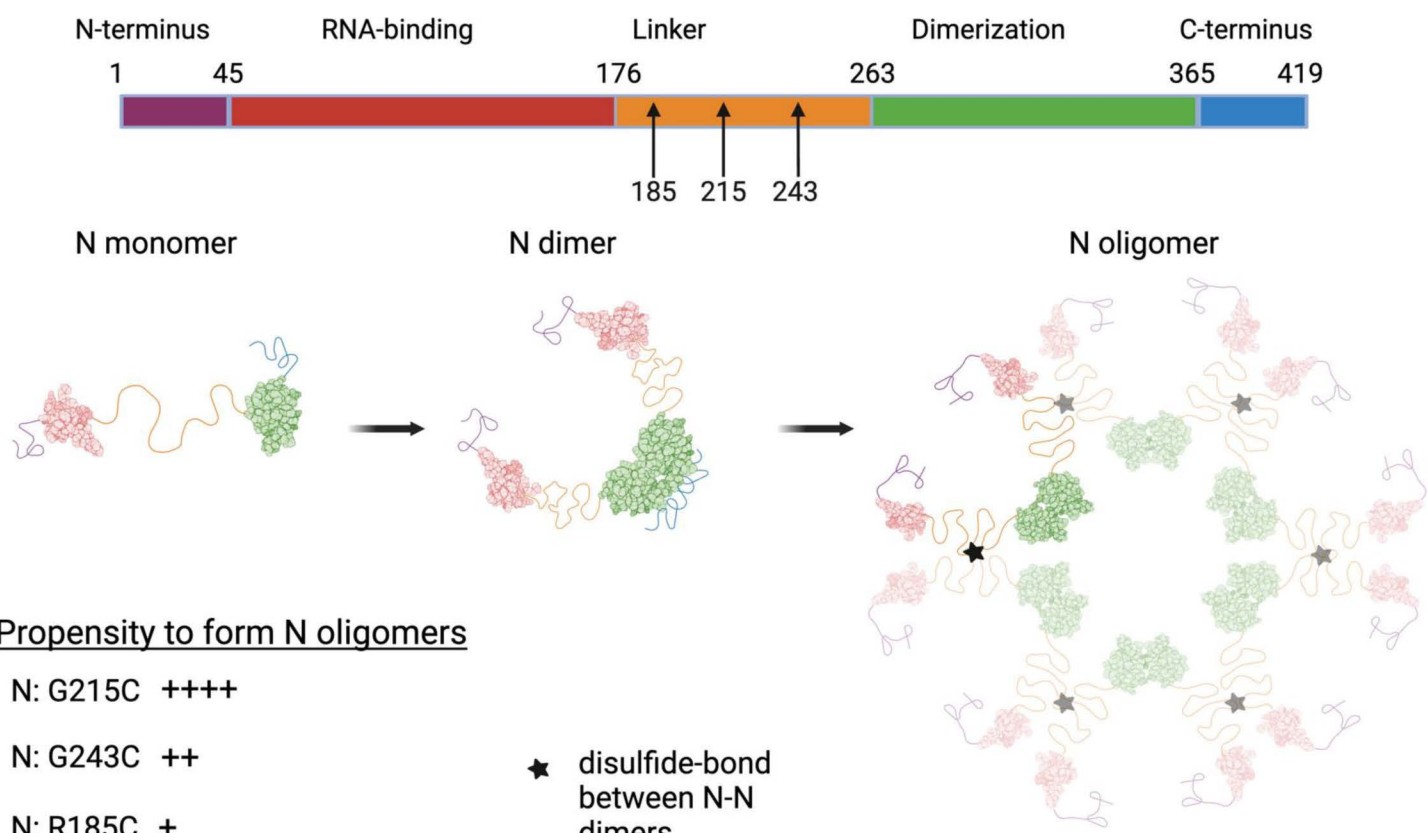

**Fig 8. Schematic of how mutations in the nucleocapsid linker may alter the oligomerization status.** The SARS-CoV-2 N protein is composed of RNA-binding (red) and dimerization domains (green), interspersed with flexible unstructured regions at the N and C-termini and a linker region (yellow) in the middle of the protein. Mutations at three separate sites in the central linker region introduce novel cysteines that differentially increase dimerization levels via a disulfide bond. (RNA-binding: PDB code 6YI3 [80], Dimerization Domain PDB code: 6WZO [9]). We propose a conceptual framework that these interactions occur between the linker regions of pairs of N-N dimers and mediate different levels of higher order N-N oligomerization. This figure was made using BioRender.

observations that the virions of a clinical Delta isolate retain the spherical shape and radius size found in the ancestral SARS-CoV-2 lineage [51], suggesting that additional mutations in the Delta lineage have the effect of counteracting the phenotype we observe with the G215C virus.

Relatedly, one of the critical unanswered questions of this study is why no Omicron sub-lineages have possessed such a cysteine in the linker region of the nucleocapsid. It is clear from human transmission data, as well as the clear beneficial effect in vitro and in vivo (Figs 4E, 5B, and 5C) of the G215C mutation that a cysteine in this region of the nucleocapsid is beneficial. Differing stability of dimer formation at different positions however (greatest at 215>243>185; Figs 2B and 3B) suggests that the 215 location is preferred over bond formation closer to either the RNA-binding or dimerization-domains. Furthermore, while the mutations at 214 and 215 were evolutionarily maintained in the Lambda

and Delta lineages, the mutations at 243 and 185 we observed in our Iota and Beta stocks were not maintained in these lineages. The 185 mutation in Beta, while present in our initial viral isolation and p1 stock, reverted back to an R during generation of the p2 stock, suggesting it decreased fitness. It is possible that there has not yet been sufficient time for a cysteine-causing mutation to arise at the 214/215 residue of nucleocapsid, and this mutation could occur in a future variant of concern (in the context of omicron or an as-of-yet unknown future variant). Our data suggest that this mutation would be likely to increase viral titers significantly, and mutations in this region should be monitored closely for their public health implications. It is also possible that there are epistatic interactions that mediate the beneficial effect of the N:G215C mutation that are not present in the Omicron background. While the N:G215C mutation was clearly beneficial for Delta (Fig 1), as well as ancestral SARS-CoV-2 in the absence of any other complementary mutations (Figs 4E, 5B, and 5C), future studies in the Omicron background would be needed to understand the effect this mutation would have on currently circulating virus lineages.

Curiously, the mutations we observe in this study (particularly 215) lie on the binding interface between the N linker and the NSP3 Ubl1 domain [52], and we hypothesize that stably dimerized N would have a reduced ability to interact with NSP3. There is clear evidence that the interaction between N and NSP3 at this binding interface is important for coronavirus biology, and mismatches between the N linker region and the NSP3 Ubl1 domain severely attenuate viral titers or block viral rescue [53,54]. The function of the N-NSP3 interaction has not been fully resolved, though it has been suggested to play critical roles in either tethering the incoming viral genome to the replication-transcription complex, or delivering the viral genome to the DMVs, to ensure successful early replication [53–55]. N is known to play a key role early in infection, as the gRNA of coronaviruses (unlike nearly all other positive strand RNA viruses) is only minimally infectious in the absence of N protein, suggesting that this N-NSP3 interaction is likely key for efficient replication or transcription of viral RNA [56–60].

In addition to its role in the replicase, NSP3 forms a key portion of the pore connecting the cytoplasm to the interior of the DMV where viral transcription occurs [61,62]. NSP3 comprises the outermost (cytoplasmic facing) layer of the DMV, with the Ubl1 domain at the very tip of the "prongs" that extend outwards from the pore into the cytoplasm [62]. Recent structural work proposed that the interaction between the linker region of N and the Ubl1 domain of NSP3 helps to mediate a condensation of the nucleocapsid protein prior to encapsidation [52]. In addition, this interaction is proposed to tether nucleocapsid molecules in the immediate vicinity of the DMV exit, ensuring that viral genomes are immediately coated with N as they are extruded out from the DMV pore [52]. Curiously, our work suggests a model in which stabilization of N-N interactions that occur through the linker region could reduce the ability of N to bind to NSP3, with the N-N linker associated oligomers and N-NSP3 binding scenarios being mutually exclusive (Fig 8). Further work is needed to reconcile this theory with the finding in this study that the G215C mutation *increases* encapsidation. We believe that the mutations and viruses characterized in this study may prove useful reagents for future studies studying the functional implications of the N-NSP3 interaction.

Our data raise one additional unanswered question, in that we observe a substantial advantage in viral growth of the N:G215C virus in differentiated human bronchial cells and hamsters but not in Vero E6 cells (Fig 4C versus Figs 4E, 6B, and 6C). Vero cells often fail to recapitulate the results seen in human culture systems; they lack the ability to produce and respond to IFN as well as encoding a different suite of cellular proteins, including kinases [3,5,8]. It is possible that the increased oligomerization of N with the G215C mutation drives a greater level of encapsidation that is better able to shield viral RNA from detection by RIG-I and the cellular innate immune response present in primary human lung cells and the hamster model. Alternatively, it is possible that there is a specific host factor present in differentiated lung cells, either a restriction factor that the N dimer is specifically able to evade, or a pro-viral factor that only the dimerized N can access. Future studies using the mutations and viruses characterized in this study will be focused on understanding the mechanism of the growth advantage of N:G215C in vitro and in vivo and the larger role that levels of N oligomerization play in coronavirus infection.

## Materials and methods

### Ethics statement

Research conducted in this study was reviewed and approved by the Institutional Biosafety Committee of the University of Vermont (REG202100001) and the Institutional Biosafety Committee of the University of Texas Medical Branch (UTMB). All studies in animals were conducted under a protocol approved by the UTMB Institutional Animal Care and Use Committee and complied with USDA guidelines in a laboratory accredited by the Association for Assessment and Accreditation of Laboratory Animal Care. UTMB is a registered Research Facility under the Animal Welfare Act. It has a current assurance (A3314-01) with the Office of Laboratory Animal Welfare, in compliance with NIH Policy. Procedures involving infectious SARS-CoV-2 were performed in the Galveston National Laboratory ABSL3 facility and the University of Vermont BSL3 facility. The use of deidentified positive clinical specimens was approved by the University of Washington Institutional Review Board (STUDY00000408) and the University of Vermont Institutional Review Board (STUDY00000881) with a consent waiver.

### Cell culture

**Stable cell lines.** Human embryonic kidney cells (HEK-293T/17) (CRL-11268, American Type culture Collection, Manassas, V) were kindly provided by J. Salogiannis (UVM); Vero-E6 cells expressing TMPRSS2 were obtained from the Japanese Cancer Research Resources Bank (JCRB1819). All cell lines were maintained in Dulbecco's Modified Eagle Medium (DMEM) (10-017-CM, Corning) and supplemented with 10% fetal bovine serum (FBS) (16140-071, Gibco). Cells were grown at 37°C and 5% $CO_2$.

**Primary cells.** Twenty-four-well transwell inserts (3470-Clear, Costar) were coated with 125 µL of a 50-µg/mL rat tail collagen (A10483-01, Gibco) in 0.02N acetic acid for 1 h at room temperature. Collagen solution was aspirated and washed once with DMEM (10-017-CM, Corning) before membranes were equilibrated with DMEM in both the apical and basolateral chambers for 1 h at 37°C. DMEM was replaced with "basal media" containing PenumaCult Ex-Plus medium (05040, Stemcell Technologies), 10 mL of PneumaCult 50× supplement (05042, StemCell Technologies), 0.001% hydrocortisone (07926, StemCell Technologies), 1% penicillin/streptomycin (30-001-Cl, Corning), 30-µg/mL gentamicin (15750060, Gibco), and 15 µg/mL amphotericin (30-003-CF, Corning) for 20 min at 37°C. Expanded human bronchial epithelia cells (CC-2541, Lonza) were plated onto transwell inserts at a density of 40,000 cells/cm² in warmed basal media. Apical media was changed the next day to remove residual DMSO after which apical and basolateral media was changed every other day until cells were 98%–100% confluent. Once at confluency, basal media was aspirated from the apical side and the basolateral chamber media was replaced with PneumaCult-ALI S media (05002, StemCell Technologies), 0.002% heparin (07980, StemCell Technologies), 0.001% hydrocortisone, 1% penicillin/streptomycin, 30-µg/mL gentamicin, 10% v/v PneumaCult-ALI supplement (05003, StemCell Technologies), 1% v/v PneumaCult-ALI maintenance Supplement (05006 StemCell Technologies), and 15 µg/mL amphotericin. Basolateral chamber medium was changed every other day until differentiation occurred, ~21 days post airlift.

### Viruses

**Viral isolation.** SARS-CoV-2 WA1/2020 was obtained from the World Reference Center for Emerging Viruses and Arboviruses at the UTMB. Viral isolates for Alpha (hCoV-19_USA_WA-UWJ17_2021, B.1.1.7.3), Beta (hCoV-19_USA_UWJ01_2021, B.1.351), Gamma (hCoV-19_USA_WA-UWJ05_2021, P.1.17/B.1.1.28.1.17), Delta (hCoV-19_USA_WA-CDC-UW21061761110_2021, AY.44; hCoV-19_USA_WA-UWD03_2021, AY.120.1; hCoV-19_USA_WA-UWD05_2021, AY.3), Epsilon (hCoV-19_USA_WA-UWJ13_2021, B.1.429), Iota (hCoV-19_USA_VT-UVM5927_2021, B.1.526), Mu (hCoV-19_USA_WA-UWM04_2021, B.1.621.1.2/BB.2), and Omicron (hCoV-19/USA/WA-CDC-UW21120651352/2021, BA.1) SARS-CoV-2 variants were obtained from primary clinical specimens. Deidentified nasal

swabs were acquired from persons who tested positive for COVID-19 and the variant lineage was determined via Next Generation Sequencing (NGS) or digital droplet polymerase-chain reaction (PCR) either before or after isolation [8]. All isolated specimens were less than eight days old before −80°C storage, stored in saline solution, and had a diagnostic PCR cycle threshold of less than 32. Virus from clinical specimens was isolated in VeroE6-TMPRSS2 cells. Cells were monitored daily for cytopathic effect (CPE) and harvested when ~50% of the cells exhibited CPE or death. Clarified samples were stored at −80°C and used to generate working viral stocks. Viral stocks were titered by focus forming assay. NGS of viral stocks was conducted by the Microbial Genome Sequencing Center or SeqCoast Genomics.

**Construction of recombinant SARS-CoV-2.** The sequence of recombinant wild-type (WT) SARS-CoV-2 is based on the USA-WA1/2020 strain provided by the World Reference Center for Emerging Viruses and Arboviruses (WRCEVA) and originally isolated by the USA Centers for Disease Control and Prevention [63]. Recombinant WT SARS-CoV-2 and mutant viruses were created using a cDNA clone and standard cloning techniques as described previously [32,64]. Construction of WT SARS-CoV-2 and mutant viruses were approved by the UTMB Biosafety Committee.

### In vitro SARS-CoV-2 infection

**Vero-TMPRSS2 cells.** Vero-TMPRSS2 cells were seeded in a 24-well plate at 50,000 cells/well 1 day before infection. The following day, cells were inoculated in 100 μL of virus at the indicated MOI for 1 h at 37°C and 5% $CO_2$, after which the inoculum was removed and replaced with fresh growth media. Supernatants were collected at the indicated timepoints, clarified to remove cellular debris, and stored at −80°C until time of titering via focus assay.

**Human bronchial epithelial cells.** Differentiated HBECs cells were washed three times with ~200 μL of 37°C HEPES buffered saline before being infected with 100 μL of virus at an MOI of 0.5 resuspended in ExPneumaCult ALI-S media + supplements (see primary cell maintenance section). After a 1-h incubation at 37°C and 5% $CO_2$, supernatant was removed. The apical side of the cells were washed with 150-μL ExPneumaCult ALI-S media + supplements at the indicated times post infection to collect virus. Viral washes were stored at −80°C until time of titering via focus assay. Basal media was changed every 48 h during the infection time course.

### Focus forming assay

Viral titrations were performed largely as previously described [65]. VeroE6-TMPRSS2 cells were seeded at a density of 60,000 cells/well into white 96-well plates (Falcon; #353296), and 24 h later, viral samples of interest were serially diluted 10-fold in DMEM + 10% FBS. Plates were aspirated and infected with diluted viral samples for 1 h at 37°C and 5% $CO_2$ before being overlaid with a 1.2% methylcellulose (Acros; #332620010) solution suspended in DMEM. After 24 h, the methylcellulose solution was aspirated, and the plates were fixed a 4% formaldehyde solution (Honeywell; #F1635-4L) for 20 min before being washed once with deionized water. Cells were permeabilized in 50 μL of 0.05% Triton X100 (Fisher; #BP151-100) for 5 min, washed with PBS, and blocked (5% non-fat milk solution in PBS) for 1 h at room temperature. Virally infected cells were detected with an anti-SARS-CoV-2 N antibody (Sinobiological; #40143-R001, 1:20,000) resuspended in 5% milk for 1 h at 37°C. Cells were washed twice with PBS and stained with a HRP-conjugated secondary antibody (Seracare; #5220-0337, 1:4,000) resuspended in 5% milk for 1 h at 37°C. Cells were washed twice with PBS, and foci developed using a TruBlue HRP substrate (SeraCare; #5510-0030). Foci were imaged on a BioTek ImmunoSpot S6 MACRO Analyzer and manually counted.

### Viral concentration

Viral stocks were concentrated by vortexing 1 mL of high titer viral stocks with 10% PEG (Sigma-Aldrich; #P6667) for ~5–10 min before centrifugation at 10,000$g$ for 30 min at 4°C. Pellets were then resuspended in an NP-40, 1% Trition X-100 lysis buffer (see SDS-PAGE and western blotting section below).

## Analysis of publicly available genomic data

The identity of the amino acid residue in the nucleocapsid protein (including at 215) in 7,342,041 SARS-CoV-2 sequences deposited in GenBank, the China National Center for Bioinformation, and from COG-UK was analyzed using the phylogenetic tree (Cov2Tree) and visualized by Taxonium on January 10th, 2024 [66–68]. Sequences for non-SARS-CoV-2 coronavirus nucleocapsid sequences were obtained through the NIH Protein Databank and aligned using Clustal Omega (EMBL-EBI, Clustal 0 (1.2.4)) [69].

## Hamster infection studies

For in vivo studies, three- to four-wk-old male hamsters were purchased from Envigio, and all studies conducted within the Galveston National Laboratory ABSL3 facility. Studies were conducted in accordance with a protocol approved by the UTMB Institutional Animal Care and Use Committee and comply with the United States Department of Agriculture guidelines. All laboratories were accredited by the Association for Assessment and Accreditation of Laboratory Animal Care. For infection studies, animals housed in groups of five were intranasally infected PBS alone (mock) or with $10^4$ FFU of WT or N:G215C mutant viruses. Animals were then monitored daily for weight loss and development of clinical disease until completion of the experiment. All procedures were carried out under anesthesia with isoflurane (Henry Schein Animal Health), except for weighing.

## Histology

Hematoxylin- and eosin-stained microscopic slides of lung from each hamster were separated into three groups according to the number of days after intranasal inoculation of SARS CoV-2. Microscopic slides of each group were scrambled thoroughly and examined blindly to assess the severity of pathologic lesions. Slides were placed in the order of serial pairwise comparison of the extent of lesions. Upon completion of the ordering of severity, numbers were assigned to the slides from 1, least pathologic change, to the highest number for the slide with the most severe pathology. At this point, the code was revealed, and the sum of the rank order of severity numbers for each group: mock, wild type, and mutant was calculated and divided according to the number of slides in the group to determine a severity score for the group. The number of slides in the groups varied slightly as tissues which contained abundant polymorphonuclear leukocytes, very likely indicating superimposed bacterial bronchopneumonia were removed and not scored. Separate sums of rank order scores and average severity score were calculated for each day.

## Antigen staining

Antigen staining was performed as previously described [70]. Briefly, cut sections were deparaffinized by heading at 56°C for 10 min, followed by 3 washes with xylene and 4 washes with graded ethanol. Slides were then rehydrated with distilled water and antigen retrieval performed by steaming slides for 40 min in antigen retrieval solution (10-mM sodium citrate, 0.05% Tween-20, ph6). After cooling, slides were rinsed with water, and endogenous peroxidases quenched in TBS + 0.3% hydrogen peroxide. Sections were then washed with TBST, blocked with 10% goat serum diluted in 1% BSA/TBST, followed by probing with primary anti-N antibody (Sino #40143-R001) at 1:1,000. Sections were then washed in three times with TBST and incubated in secondary HRP-conjugated anti-rabbit antibody (Cell Signaling Technology #7074) at 1:200. To visualize antigen, sections were treated with ImmPACT NovaRED (Vector Laboratories #SK-4805) for 3 min. Sections were then counter stained with hematoxylin for 5 min. Finally, sections were the dehydrated by incubation in graded ethanol followed by xylene and mounted with cover slips. Viral antigen staining occurred through blinded scoring on a scale of 0–4, with the scores of two lung sections being averaged to determine the final score.

## Plasmid generation

The sequences for the N protein of our Beta, Delta, and Iota SARS-CoV-2 variants, as well as the WA1/2020 ("WT") sequences were synthesized and subcloned into a pcDNA3.1 vector (GenScript). In addition, point mutants were

generated in which the cysteine in the Beta, Iota, and Delta N sequences was mutated back to the corresponding residue in WA1 (Beta: C185R, Delta: C215G, Iota:C243G) or where cysteine were introduced at the 185, 215, or 243 positions into the WA1 background (R185C, G215C, G243C). Plasmid sequences were confirmed via whole-plasmid sequencing (Plasmidsaurus).

## Transfection

HEK-293T cells were seeded at a density of 350,000 cells/well in a 12-well plate (Corning #3512). The following day, 500 ng of the appropriate plasmid and 2 µL of Lipofectamine (Invitrogen #52887) diluted to a volume of 40 µL in DMEM were incubated for 24 h at 37°C and 5% $CO_2$. Cells were harvested by scraping into PBS, pelleted, and lysed for SDS-PAGE and western blotting (see below).

## SDS-PAGE and western blotting

Cells were lysed in an NP-40 lysis buffer (Thermo Scientific #J60766.AK) containing a 1% Triton X-100 solution (Fisher, #BP151-100) and protease inhibitors (Thermo Scientific Fisher #A32955) for 20 min on ice. Cell lysates were clarified by spinning at 14,000 rpm for 10 min at 4°C to remove insoluble debris and diluted in a 1:1 ratio (v/v) of 4× Laemmli sample buffer (250 mM Tris-HCl pH 6.8, 40% glycerol, 8% SDS, 0.04% Bromophenol Blue). Reduced lysates were treated with either 2.75-mM 2-mercaptoethanol (Gibco #21985-023) or 10-mM dithiothreitol (Fisher Bioreagents # BP172-5). For NEM processing, samples were lysed in a buffer containing 200-U/mL catalase (to quench $H_2O_2$), 1-mM DPC-Bio 1 (to block sulfenylated proteins/-SOH), and increasing concentrations of NEM (1–100 mM) (NEM; Thermo Fisher #23030) according to previously validated protocols designed to prevent the formation of post-lysis disulfide bonds [71,72]. Samples that received NEM pre-treatment were incubated for 30 min at 37ºC NEM at the indicated concentration in PBS (pH 7.4) before lysis in the buffer described above (fresh NEM present in the lysis buffer). All samples were boiled for 5 min before loading. Samples were separated in NuPAGE 4%–12% Bis-Tris gels (Invitrogen # NPO335BOX) in MES buffer (Invitrogen # NP0002) with a molecular mass ladder (Thermo Fisher #LC5925) at 180 V for 50 mins before being transferred into a nitrocellulose membrane (Invitrogen # IB23001) using an iBlot 2 machine (Invitrogen) at 20 V for 7 min.

Membranes were blocked in a 5% milk/PBS solution for 30 min and then incubated overnight at 4°C in a solution containing the primary antibody in a 5% milk/PBST (0.2% Tween 20 (Thermo Fisher Scientific, # J20605-AP). Membranes were washed in PBST, incubated while rocking with secondary antibodies diluted in 5% milk/PBST for 45 min, washed again with PBST, and imaged with a LI-COR Odyssey CLx. Protein expression was analyzed by measuring band densitometry in Fiji (22743772) (Table 1, Antibodies).

**Table 1. Antibodies.**

| Primary Antibodies | | |
|---|---|---|
| **Species, Target** | **Catalog #, Company** | **Dilution Used** |
| SARS-CoV-2 nucleocapsid protein (mouse) | Invitrogen, MA5-35943 | 1:1,000 |
| SARS-CoV-2 nucleocapsid protein (rabbit) | SinoBiological, 40143-R001 | 1:10,000 (western) 1:20,000 (focus assay) |
| SARS-CoV-2 membrane (M) protein (mouse) | Cell Signaling, E5A8A | 1:1,000 |
| β-actin (mouse) | Novus Biologicals, NB600−501 | 1:10,000 |
| **Secondary Antibodies** | | |
| IRDye 680LT goat anti-rabbit IgG | LICOR, #92668021 | 1:10,000 |
| IRDye 680RD goat anti-mouse IgG | LICOR, #92668070 | 1:10,000 |
| IRDye 800CW goat anti-mouse IgG | LICOR, #92632210 | 1:10,000 |
| HRP anti-rabbit | Seracare; #5220-0337 | 1:4,000 |

## Immunoprecipitation assays

VeroE6-TMPRSS2 cells were infected at an MOI of 0.01 with WT (WA1) or Delta SARS-CoV-2 for proteomic experiments, and 24 hpi cells were scraped into PBS, pelleted, and lysed in NP-40/Triton lysis buffer (see SDS-PAGE and western blotting). Alternatively for immunoprecipitations for RT-qPCR, viral supernatants from PEG concentrated samples were used. SARS-CoV-2 N was affinity purified using 10 µL of an anti-SARS-CoV-2 N antibody (40143-R001, Sinobiological) and Protein G-conjugated Dynabeads (Invitrogen, #10003D) according to the manufacturer instructions.

## RNA extraction and detection by RT-qPCR

Viral RNA was extracted from immunoprecipitated samples by placing magnetic beads in RLT buffer and performing an RNA extraction using the QIAamp Viral RNA mini kit (Qiagen, Cat. No. 52904) according to manufacturer's instructions, as previously described [73]. SARS-CoV-2 viral RNA (no strand specificity) was detected using primer-probe set N1 from IDT's 2019-nCoV CDC Emergency Use Authorization Kits, as previously described [73].

## Mass spectrometry

After immunoprecipitation, samples were run on an 8% BOLT gel (Invitrogen, # NW00085BOX), and bands were visualized via Coomassie stain (40% methanol, 20% acetic acid, 0.01% Brilliant Blue) and de-stained (30% methanol, 10% acetic acid). Bands of interest (monomer, dimer, or full lane) were cut out of the gel, and further cut into ~1-mm cubes before being placed into tubes for processing using HPLC-grade Fisher brand chemicals. Gel slices were incubated in water for 5 min and then de-stained for 30 min at 37°C (50-mM ammonium bicarbonate, 50% acetonitrile). Slices were rinsed in fresh acetonitrile for two 5-min incubations before incubating for 30 min at 55°C in 25-mM DTT (Thermo Fisher, #R0861)/50-mM ammonium biocarbonate. Slices were cooled, incubated in 100% acetonitrile for 5 min, and incubated for 45 min at room temperature in 10-mM iodoacetamide (Sigma-Aldrich, #I6125)/50 mM ammonium bicarbonate, in the dark. Slices were incubated in (50-mM ammonium bicarbonate, 50% acetonitrile) for 5 min, rinsed in water for 10 min, and centrifuged at 13,000 rpm for 30 s. Next, samples were dehydrated in acetonitrile before all liquid removed and gel slices allowed to dry completely. Once dry, samples were placed on ice for 5 min before incubating for 30 min on ice with 6-ng/µL trypsin (Promega #V511A) in 50-mM ammonium bicarbonate and then digesting overnight at 37°C. Supernatants were removed and saved, while gel slices were covered with a 50% acetonitrile, 2.5% formic acid solution, and spun for 10 min after which supernatants were combined with those from the previous step. Finally, 30 µL acetonitrile was added to gel slices for a final 10-min incubation, after which these supernatants were combined with those from the previous two steps and dried in a vacuum centrifuge at 37°C.

Following peptide preparation, samples were resuspended in 2.5% acetonitrile and 0.1% formic acid. Then, loaded onto an EASY n-LC 1,200 for HPLC (300 nL/min) followed by tandem MS/MS on an Exploris mass spectrometer (ThermoFisher). The Exploris was fitted with a Nanospray Flex ion source and chromatograph column packed in-house (35 cm × 100, 1.8 µm 120 Å UChrom, C18 packing material). Peptides were eluted into the mass spectrometer on a 5%–95% gradient of 80% acetonitrile and 0.1% formic acid over 170 min, followed by a 10-min clear at 5% gradient. The following were the mass spectrometer parameters. Precursor scan range = 350–1,400 $m/z$, resolution = 60,000, normalized AGC target = 250% of maximum, maximum IT = 25 ms. Data-dependent MS2 orbitrap resolution = 15,000, normalized AGC target = 50% of maximum, isolation window = ±1.4 $m/z$, collision energies = 30%, and dynamic exclusion = 45 s with repeat count of 1. Followed by 10 collision induced dissociation tandem mass spectra of the top 10 most abundance ions in the precursor scan.

Resulting raw mass spectra for each sample was searched against both a forward and reverse African Green Monkey protein database and a SARS CoV-2 protein database using SEQUEST [74]. Search parameters included a requirement for tryptic peptides and differential modifications (phosphorylation on serine, threonine, and tyrosine [+79.9663 Da],

oxidation of methionine [+15.99.49 Da], acrylamindation of cysteine [+71.0371 Da], and carboxyamidomethylation of cysteine [+57.0214 Da]). The resulting peptide lists were filtered by mass accuracy (ppm ≤ ±5), cross correlation scores ($z$=1 XCorr ≥ 1.8, $z$=2 XCorr ≥ 2, $z$=3 XCorr ≥ 2.2, $z$=4 XCorr ≥ 2.4, and $z$=5 XCorr ≥ 2.6), and a unique delta-correlation (uniqΔcorr ≥ 0.15). All filtering resulted in a false discovery rate of less than 0.01%. The filtered peptide lists were then compared by sample type (present or absent or enriched [2.5-fold increase]) by spectral count using R software.

## Sample preparation for electron microscopy

Vero-E6 cells were cultured on plates and infected with mNG WT or G215C SARS-CoV-2 under BSL-3 conditions. Cells were pre-fixed with 3% glutaraldehyde, 1% paraformaldehyde, 5% sucrose in 0.1-M sodium cacodylate, then removed from the plates, and transferred to Eppendorf tubes and gently pelleted. The fixative supernatant was removed and pellets rinsed with fresh cacodylate buffer containing 10% Ficoll, placed into brass planchettes (Ted Pella), and rapidly frozen with an HPM-010 high-pressure freezing machine (Bal-Tec, Leichtenstein). The frozen samples were transferred under liquid nitrogen to cryotubes (Nunc) containing a frozen solution of 2.5% osmium tetroxide, 0.05% uranyl acetate in acetone. Tubes were loaded into an AFS-2 freeze-substitution machine (Leica Microsystems, Vienna) and processed at −90°C for 72 h, warmed over 12 h to −20°C, held at that temperature for 6 h, and then warmed to 4°C for 1 h. Samples were rinsed 3× with cold acetone, after which they were infiltrated with Epon-Araldite resin (Electron Microscopy Sciences) over 48 h. Cell pellets were flat-embedded between two Teflon-coated glass microscope slides and the resin was polymerized at 60°C for 48 h.

## Electron microscopy and dual-axis tomography

Embedded cells were observed by light microscopy and appropriate blocks were extracted with a microsurgical scalpel and glued to the tips of plastic sectioning stubs. Semi-thin (170 nm) serial sections were cut with a UC6 ultramicrotome (Leica Microsystems) using a diamond knife (Diatome, Switzerland). Sections were placed on formvar-coated copper-rhodium slot grids (Electron Microscopy Sciences) and stained with 3% uranyl acetate and lead citrate. Gold beads (10 nm) were placed on both surfaces of the grid to serve as fiducial markers for subsequent image alignment. Grids were placed in a dual-axis tomography holder (Model 2040, E.A. Fischione Instruments) and imaged with a Tecnai T12-G2 transmission electron microscope operating at 120 KeV (ThermoFisher Scientific) equipped with a 2k × 2k CCD camera (XP1000; Gatan). Tomographic tilt-series and large-area montaged overviews were acquired automatically using the SerialEM software package [75]. For tomography, samples were tilted ±62° and images collected at 1° intervals. The grid was then rotated 90° and a similar series taken about the orthogonal axis. Tomographic data were calculated, analyzed, and modeled using the IMOD software package [76–78] on iMac Pro and Mac Studio M1 computers (Apple).

## Statistics

Unpaired T-tests on raw or log-transformed (viral titer) data were performed using GraphPad Prism 10 or R [79]. Statistical significance is indicated with· ($p$ = 0.05–0.1), * ($p$ < 0.05), ** ($p$ < 0.01), *** ($p$ < 0.001), or **** ($p$ < 0.0001).

## Supporting information

**S1 Fig. Absence of cysteines in the linker region of Coronaviridae nucleocapsid proteins.** Sequences for the indicated coronavirus nucleocapsid protein sequences were obtained through the NIH Protein Databank and aligned using Clustal Omega from EMBL-EBI (Clustal 0 (1.2.4)). The navy box highlights the linker region (residues 175−247 in SARS-CoV-2). Shown in red is the only cysteine in the displayed sequences, which occurs immediately before the linker region in Erinaceus betacoronavirus.
(TIF)

**S2 Fig. SARS-CoV-2 N dimer formation.** VeroE6-TMPRSS2 cells were infected, or mock infected, with the indicated SARS-CoV-2 variants (or WA1, termed Wild type) at an MOI of 0.01 for 24 h. **(A)** Cell lysates were harvested under reducing conditions (10-mM DTT) before visualization by SDS-PAGE and western blot using antibodies recognizing SARS-CoV-2 N (red) and β-actin (green). Alternatively, unreduced lysates from cells infected as above were visualized by SDS-PAGE and western blot using two different antibodies recognizing SARS-CoV-2 N: **(B)** Invitrogen anti-N mouse antibody (MA5-35943, in green) and **(C)** Sino Biological anti-N rabbit antibody (40143-R001 in red). **(D)** Lysates were stained and imaged simultaneously with the two anti-N antibodies and the overlay is shown in yellow. Note minor bands that are seen with the Invitrogen but not the Sino-Biological antibody. The MW of the ~100-kDa band observed in Beta, Delta, and Iota samples under non-reducing conditions is indicated by a *. A representative gel from three **(A)** or two **(C)** independent biological replicates is shown. **(E)** Vero-TMPRSS2 cells were infected with WA1 or Delta SARS-CoV-2 at an MOI of 0.01. Twenty-four hpi cells were harvested, lysed, and N was affinity purified using immunoprecipitation. N and associated proteins were run on an SDS-PAGE gel, and bands corresponding to the monomer and dimer were cut and processed for mass spectrometry. **(F)** The number of peptides for each SARS-CoV-2 viral protein found in the monomer or dimer portion of the gel, for either WT (WA1) or Delta, is shown. The data underlying this Figure can be found in S1 Data and S2 Data. (TIF)

**S3 Fig. Quantification of NEM treatment on N dimer formation. (A)** VeroE6-TMPRSS2 cells were infected with WT or the G215C virus at an MOI of 0.01 for 24 h. Cell lysates were harvested in standard triton lysis buffer (lanes 1, 5) or in the presence of increasing concentrations of N-ethylmaleimide (1, 10, 100 mM). **(B)** A parallel experiment was performed in which cells were pre-treated by incubating for 30 min at 37°C in increasing concentrations (1, 10, 100 mM) of NEM in PBS. Cells were then lysed as in **(A)**, with NEM present in the lysis buffer as well as the pre-incubation. Lysates were visualized by SDS-PAGE and western blot using antibodies to SARS-CoV-2 N and actin. **(C and D)** Quantification of the N dimer to monomer ratios in **(A and B)** is shown for two independent experiments. The data underlying this Figure can be found in S1 Data. (TIF)

**S4 Fig. The G215C mutation increases levels of RNA bound to nucleocapsid.** VeroE6-TMPRSS2 cells were infected with WT or the G215C viruses at an MOI of 0.01 for 48 h. Clarified viral supernatants were concentrated by binding to PEG and centrifuging at 10,000*G* for 30 min at 4°C. Concentrated virus was lysed and nucleocapsid proteins from each flask were affinity purified. Western blots were used to verify equal inputs of N from all conditions (inputs), and saturation of the beads (flow-through). The data underlying this Figure can be found in S1 Data. (TIF)

**S5 Fig. Antigen staining of lung tissue from infected hamsters.** Section of lung tissue from infected hamsters was stained for viral antigen (nucleocapsid). Sections were then blinded and scored on a 4 point scale for parenchyma, airway, and overall staining **(A)**. Each data point represents the average score form two lung section from each hamster in the group (*n* > 4). Horizontal lines represent the group mean and error bars are ±SD. Significance determined by student *T* test. **(B)** Representative images are shown for WT and N:215C infected animals for 2, 4, and 7 dpi. (. [*p* = 0.05−0.1], * [*p* < 0.05]). The data underlying this Figure can be found in S1 Data. (TIF)

**S6 Fig. The G215S mutation decreases packaging of N and growth in primary differentiated human bronchial cells. (A)** VeroE6-TMPRSS2 cells were infected with the WT or N:G21SC viruses at an MOI of 0.0005 for 1 h, viral supernatants were collected at 8, 24, 32, 48, and 72 h post infection and titered by focus forming assay (FFU; focus forming unit). **(B)** Human bronchial epithelial cells were infected with WT or N:G215S viruses at an MOI of 0.5 for 1 h, apical washes were collected sequentially from the same well at 8, 24, 32, 48, and 72 h post infection and titered by focus

forming assay (FFU). **(C)** High titer viral stocks of the WT or N:G215S viruses were concentrated by binding to 10% poly-ethylene glycol then centrifugated at $10,000G$ for 30 min at 4°C. Unreduced lysates were collected 24 h post transfection directly from the concentrated viral pellet and N and M were visualized by SDS-PAGE. **(D)** The ratio of N to M is shown, as well as **(E)** the ratio of N to FFU. Mean ± SEM is plotted, $N = 6$ from two biological experiments **(A)**, $N = 6$ **(B)**, and $N = 3$ **(C–E)** is shown. Statistical comparisons were conducted using a two-tailed $T$ test, (. [$p = 0.05−0.1$], * [$p < 0.05$]). Limit of detection (LoD) is 20 PFU/mL and the y-axis minimum is set as the LoD. The data underlying this Figure can be found in S1 Data.
(TIF)

**S7 Fig. N:G215C virions are enlarged.** Vero-TMPRSS2 cells were infected with WT or N:G215C viruses at an MOI of 0.1. The following day cells were prepared for electron microscopy by high-pressure freezing and freeze-substitution, then sectioned and imaged by dual-axis electron tomography. Virus-containing exit compartments were located in both samples, and 20 virions that had completely separated from cellular membranes were randomly selected and imaged for each virus.
(TIF)

**S1 Data. The individual datapoints underlying each graph can be found in this dataset.**
(XLSX)

**S2 Data. The proteolytic peptides identified in the mass spectrometry experiment in S2 Fig can be found in this dataset.**
(XLSX)

**S1 Raw Images. The uncropped western blots shown in figures and quantifications can be found in this dataset.**
(PDF)

## Acknowledgments

We would like to thank Drs Matthew Poynter, Brian Cunniff, Nate Shannon, John Salogiannis, Dimitry Krementsov, and Joyce Oetjen for technical assistance.

The authors thank the Caltech Beckman Institute CryoEM Facility for use and maintenance of the Tecnai T12 TEM.

## Author contributions

**Conceptualization:** Hannah C. Kubinski, Hannah W. Despres, Emily A. Bruce.

**Formal analysis:** Hannah C. Kubinski, Hannah W. Despres, Bryan A. Johnson, Caroline M. Dumas, David J. Shirley, Bryan A. Ballif, Vineet D. Menachery, Emily A. Bruce.

**Funding acquisition:** Vineet D. Menachery, Emily A. Bruce.

**Investigation:** Hannah C. Kubinski, Hannah W. Despres, Bryan A. Johnson, Madaline M. Schmidt, Sara A. Jaffrani, Allyson H. Turner, Conor D. Fanuele, Kumari G. Lokugamage, Caroline M. Dumas, Leah K. Estes, Bruno Martorelli Di Genova, David H. Walker, Bryan A. Ballif, Mark S. Ladinsky, Emily A. Bruce.

**Methodology:** Emily A. Bruce.

**Project administration:** Emily A. Bruce.

**Resources:** Margaret G. Mills, Andrew Pekosz, Jessica W. Crothers, Pavitra Roychoudhury, Alexander L. Greninger, Keith R. Jerome, Vineet D. Menachery, Emily A. Bruce.

**Supervision:** Bryan A. Ballif, Pamela J. Bjorkman, Vineet D. Menachery, Emily A. Bruce.

**Validation:** Emily A. Bruce.

**Visualization:** Hannah C. Kubinski, Hannah W. Despres, Bryan A. Johnson, Mark S. Ladinsky, Emily A. Bruce.

**Writing – original draft:** Hannah C. Kubinski, Hannah W. Despres, Emily A. Bruce.

**Writing – review & editing:** Hannah C. Kubinski, Hannah W. Despres, Bryan A. Johnson, Madaline M. Schmidt, Sara A. Jaffrani, Allyson H. Turner, Conor D. Fanuele, Margaret G. Mills, Kumari G. Lokugamage, Caroline M. Dumas, David J. Shirley, Leah K. Estes, Andrew Pekosz, Jessica W. Crothers, Pavitra Roychoudhury, Alexander L. Greninger, Keith R. Jerome, Bruno Martorelli Di Genova, David H. Walker, Bryan A. Ballif, Mark S. Ladinsky, Pamela J. Bjorkman, Vineet D. Menachery, Emily A. Bruce.

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
