## [Editor Report · Decision Letter 0]

21 Feb 2025

Dear Emily, 

Thank you for submitting your manuscript entitled "Variant mutation G215C in SARS-CoV-2 nucleocapsid enhances viral infection via altered genomic encapsidation" for consideration as a Research Article by PLOS Biology.

We are trying to get the confirmation from Nature Communications that your submission was there and the reviewer identities. We will try not to go to another round of review, but it would be good to get all the information possible. 

To further continue with the process, we need you to complete your submission by providing the metadata that is required for full assessment. To this end, please login to Editorial Manager where you will find the paper in the 'Submissions Needing Revisions' folder on your homepage. Please click 'Revise Submission' from the Action Links and complete all additional questions in the submission questionnaire. 

For the Editorial Manager team, please edit your cover letter to let them know about the previous review and include the name of the journal where the work was previously considered and the manuscript ID it was given. In addition, please upload a response to the reviews as a 'Prior Peer Review' file type, which should include the reports in full and a point-by-point reply detailing how you have or plan to address the reviewers' concerns, this should include all rounds of review and e-mail exchanges with the editor of NatComms. 

Once your full submission is complete, your paper will undergo a series of checks in preparation for peer review. After your manuscript has passed the checks it will be sent out for review. To provide the metadata for your submission, please Login to Editorial Manager (https://www.editorialmanager.com/pbiology) within two working days, i.e. by Feb 23 2025 11:59PM.

Kind regards,

Melissa

Melissa Vazquez Hernandez, Ph.D.

Associate Editor

PLOS Biology

---

## [Editor Report · Decision Letter 1]

1 Mar 2025

Dear Emily,

Thank you for your patience while we considered your revised manuscript "Variant mutation G215C in SARS-CoV-2 nucleocapsid enhances viral infection via altered genomic encapsidation" for publication as a Research Article at PLOS Biology. This revised version of your manuscript has been evaluated by the PLOS Biology editors and Academic Editor with relevant expertise.

Based on our Academic Editor's assessment of your revision, we are likely to accept this manuscript for publication, provided you satisfactorily address the remaining editorial requests. Please also make sure to address the following data and other policy-related requests.

a) We would strongly recommend that you deposit your plasmids to Addgene to comply with our Open Science values

Please supply the numerical values either in the a supplementary file or as a permanent DOI’d deposition for the following figures:

Figure 2B, 3BD, 4CE, 5BCDE, 6ABCEFG, 7B, S2CD, S5A, S6ABDE

c) Please cite the location of the data clearly in all relevant main and supplementary Figure legends, e.g. “The data underlying this Figure can be found in S1 Data” or “The data underlying this Figure can be found in https://doi.org/10.5281/zenodo.XXXXX”

d) We require the original, uncropped and minimally adjusted images supporting all blot and gel results reported in the Figures 2ACD, 3AC, 4BD, 6D, S2ABCDE, S2AB, S4, S6C.

We will require these files before a manuscript can be accepted so please prepare and upload them now. Please carefully read our guidelines for how to prepare and upload this data: https://journals.plos.org/plosbiology/s/figures#loc-blot-and-gel-reporting-requirements

e) Please provide the tree files for the phylogenetic trees in Figures 1A

f) this is also not necessary, but we would like to encourage that if you have any additional microscopy pictures related to Figures 5F, 7A, S5B, S7, you can upload them to repositories like Zenodo

g) Please ensure that your Data Statement in the submission system accurately describes where your data can be found and is in final format, as it will be published as written there.

h) Per journal policy, if you have generated any custom code during the course of this investigation, please make it available without restrictions upon publication. Please ensure that the code is sufficiently well documented and reusable, and that your Data Statement in the Editorial Manager submission system accurately describes where your code can be found.

We expect to receive your revised manuscript within two weeks. 

*Published Peer Review History*

*Press*

Sincerely,

Melissa

Melissa Vazquez Hernandez, Ph.D.

Associate Editor

PLOS Biology

---

## [Editor Report · Decision Letter 2]

12 Mar 2025

Dear Emily,

Thank you for the submission of your revised Research Article "Variant mutation G215C in SARS-CoV-2 nucleocapsid enhances viral infection via altered genomic encapsidation" for publication in PLOS Biology. On behalf of my colleagues and the Academic Editor, Andy Mehle, I am pleased to say that we can in principle accept your manuscript for publication, provided you address any remaining formatting and reporting issues. These will be detailed in an email you should receive within 2-3 business days from our colleagues in the journal operations team; no action is required from you until then. Please note that we will not be able to formally accept your manuscript and schedule it for publication until you have completed any requested changes.

PRESS

Sincerely, 

Melissa

Melissa Vazquez Hernandez, Ph.D., Ph.D.

Associate Editor

PLOS Biology
